# Waveguide holography for 3D augmented reality glasses

Changwon Jang[1,3] ✉, Kiseung Bang[1,3], Minseok Chae [2], Byoungho Lee [2] &
Douglas Lanman[1]

Near-eye displays are fundamental technology in the next generation computing platforms for augmented reality and virtual reality. However, there are remaining challenges to deliver immersive and comfortable visual experiences to users, such as compact form factor, solving vergence-accommodation conflict, and achieving a high resolution with a large eyebox. Here we show a compact holographic near-eye display concept that combines the advantages of waveguide displays and holographic displays to overcome the challenges towards true 3D holographic augmented reality glasses. By modeling the coherent light interactions and propagation via the waveguide combiner, we demonstrate controlling the output wavefront using a spatial light modulator located at the input coupler side. The proposed method enables 3D holographic displays via exit-pupil expanding waveguide combiners, providing a large software-steerable eyebox. It also offers additional advantages such as resolution enhancement capability by suppressing phase discontinuities caused by pupil replication process. We build prototypes to verify the concept with experimental results and conclude the paper with discussion.

Near-eye display technology is evolving rapidly along with the pursuit of next generation computing platforms. For augmented reality (AR)[1] in particular, various see-through near-eye display architectures have been invented and explored in the recent decades. Examples include birdbath type displays, curved mirror type displays, retinal projection displays, and pin mirror displays[2–4]. Among the plethora of architectures, waveguide image combiners (or exit-pupil expanding waveguides) remain a leading candidate for augmented reality glasses in the industry because of their compact form factor[5,6]. Additionally, there has been significant effort to realize 3D holographic displays that provide realistic visual experiences[7]. In this work, we propose a display architecture that combines the advantages of both waveguide displays and holographic displays, enabling the path towards true 3D holographic AR glasses.

As a near-eye display application, the waveguide image combiner or waveguide display refers to a thin, transparent slab that guides the light as a total internal reflection (TIR) mode and replicates the exit-pupils to be delivered to the user's eye. These waveguides can be designed using different types of light coupling elements. Geometric waveguides use partially reflective surfaces inside the slab to re-direct and extract the light from the waveguide[5,8–10]. Diffractive waveguides may utilize surface relief gratings, volume Bragg gratings, polarization gratings, and meta surface or geometric phase elements as in/out-couplers[11–13]. TIR propagation allows the optical path to be secured in the waveguide without being obstructed, while no bulky projector or imaging optics are needed to be placed in front of user's eye. The image projector of a waveguide display is typically located at the temple side with an infinity corrected lens, providing high resolution images. The most unique advantage of a waveguide is its étendue expansion capability by pupil replications[14]. This provides a sufficient eyebox with a fairly large field of view while many other architectures suffer from their trade-off relation imposed by limited étendue. Such advantages make waveguide displays the leading technology of AR displays in recent years[4].

Despite the advantages of waveguide displays, there are some limitations to be addressed. First, waveguides can only convey a fixed

---

[1]Reality Labs Research, Meta, Redmond, WA, USA. [2]Seoul National University, Seoul, Republic of Korea. [3]These authors contributed equally: Changwon Jang, Kiseung Bang. ✉e-mail: changwon.jang@meta.com

depth, typically as infinity conjugate images. If finite-conjugate images are projected into the waveguide, the pupil replication process produces copies of different optical paths and aberrations that create severe ghost noise, which is often called *focus spread effect*[6]. Generating natural focus cues and addressing the vergence-accommodation conflict[7,15] are among the challenging goals of AR in the pursuit of realistic and comfortable visual experiences. Dual or multi-imaging plane waveguide architectures have been studied[16,17], but inherently lead to a bulkier form factor and diminished performance, along with added hardware restrictions. Additionally, achieving sufficient brightness with conventional light sources, such as micro LEDs[18], is challenging due to the low efficiency of waveguide image combiners. Although laser light sources could greatly reduce the loss from coupling efficiency, their use with waveguides is restricted because coherent light interaction during TIR propagation leads to artifacts and significant image quality degradation.

Meanwhile, holographic display technology is believed to be the ultimate 3D display approach, which modulates the wavefront of light using spatial light modulators (SLMs)[7]. It also offers unique benefits such as aberration-free, high-resolution images, per-pixel depth control, ocular parallax depth cues, vision correction functionality[19–21], as well as a large color gamut. Recently, a lot of progress has been made in the field of computer-generated hologram (CGH) rendering, garnering more attention from the industry[22–36]. Several conventional issues with holographic displays, including speckle, image quality, and heavy computational load, have been shown to be resolved with the help of enhanced CGH rendering models and the increased computing power of recent graphics processing units (GPUs)[37–40]. However, designing a compact architecture for near-eye holographic displays remains an unsolved problem due to limited étendue[27,41]. Retinal projection type designs have been explored with a holographic projector at the temple side that projects the hologram via oblique free-space projection to the eyepiece combiner[22,42,43]. However, such configurations have limited space and angular bandwidth to transmit enough étendue from the temple side to the eyepiece even with mechanical pupil steering[42,43], making the ergonomic glasses form factor an even more ambitious goal.

There have been early efforts to use waveguides as an illumination source to produce a projection pattern or image formed by an out-coupler grating with an embedded hologram pattern[44–46]. Because only a static image could be displayed and no information was carried inside the waveguide until out-coupled, this approach was not suitable for augmented reality display purposes, but represented a very early stage attempt to combine waveguides and holograms together.

Recently, researchers have attempted to implement dynamic holographic displays using the light guiding slabs[47], with further efforts being made to compensate for aberrations and improve image quality[48–50]. While they share similar motivations for transmitting holograms via waveguides, there are fundamental limitations on scalability because the method is not intended to support pupil replication; in other words, the focus spread effect remains unsolved. The light guiding slab must be thick enough to avoid replication, otherwise the overlapped wavefront becomes scrambled, creating severe artifacts such as multiple ghost images and low contrast. As a result, thick substrates ($3^{47}$–$8^{49}$ mm) are chosen for such architectures, which would not be suitable for true glasses form factor. Additionally, the eyebox and field of view are fundamentally limited to be small in such architectures[6].

In this study, we present a compact near-eye display system titled *waveguide holography*, which combines the merits of waveguide image combiners and holographic displays. Our approach fundamentally differs from previous works[47–50] as it addresses the focus spread effect of exit-pupil expanding waveguides. The core idea is to model the coherent light interaction inside exit-pupil expanding waveguides as a propagation with multi-channel kernels. Precise model calibration is

enabled by a complex wavefront capturing system and algorithm based on phase-shifting digital holography. As a result, we demonstrate that the out-coupled wavefront from the waveguide can be precisely controlled by modulating the input wavefront using our model.

We experimentally verify the capability of displaying full 3D images and the étendue expansion, which enables a large software-steered eyebox. In addition, we demonstrate that our method offers enhanced resolution beyond the limit of conventional waveguide displays. We present a detailed analysis of architecture design and scalability in the Supplementary Material (See Supplementary Figs. 4-8), and conclude in the Discussion section with some limitations as well as interesting future works.

## Results

### Architecture
Figure 1a illustrates the architecture of the proposed system, while Fig. 1b, c illustrate the compact prototype and benchtop prototype, respectively. The system consists of a collimated laser light source, a spatial light modulator (SLM), a exit-pupil expanding waveguide with surface relief gratings, and linear polarizers laminated on the SLM and out-coupler of the waveguide. Compared with the conventional waveguide display, the major difference is that the image projector is replaced with the hologram projection module. The SLM is placed without any projection lens, eliminating the need of physical propagation distance, as well as achieving a light weight design. The benchtop prototype is built on the optical table with the same architecture and similar specifications, while the SLM is relayed with de-magnifying 4-$f$ imaging system. The benchtop prototype is useful for iterating design parameters and benchmarking the performance, while the compact prototype showcases its form factor. More details and further miniaturization methods are provided in Method section.

The input light is modulated by the SLM and coupled by the in-coupler grating into the waveguide. The light propagates as a total internal reflection mode and is diffracted by an exit-pupil expanding (EPE) grating and out-coupler grating that are typically designed as leaky gratings[14]. This pupil-replication process generates manifold shifted copies of the wavefront having different optical paths inside the waveguide, that interfere with each other so that the phase and intensity of the final output wavefront is intricately scrambled. In conventional waveguide image combiners, these phenomena are understood as coherence artifacts which should be avoided. However, we fully exploit this coherent interaction of light to precisely shape the output wavefront using spatial light modulator from the hologram projection module.

Note that the étendue of transmitted light is expanded by the exit-pupil expanding waveguide, but the bandwidth of information is unchanged. Thus, controlling the entire output wavefront from modulating only the input wavefront is fundamentally an over-constrained problem. To overcome the shortage of information bandwidth, we take advantage of the fact that most of the output wavefront does not enter the eye pupil. We set the virtual target aperture at the eyebox domain as a region of interest (ROI) for wavefront shaping, and this aperture can be computationally steered to match the size and 3D location of user's eye pupil. This idea is similar to some of the previous CGH generation algorithms[51]. With the aid of eye tracking, the system can fully utilize the expanded étendue and achieve a software-steered eyebox without mechanical steering as large as conventional waveguide displays can provide.

### Modeling of hologram propagation in waveguides
In other to model the coherent light interaction inside the exit-pupil expanding waveguide, we start with making an assumption that the waveguide can be approximated as a linear shift invariant (LSI) system. The light in-coupling and out-coupling process of the waveguide can

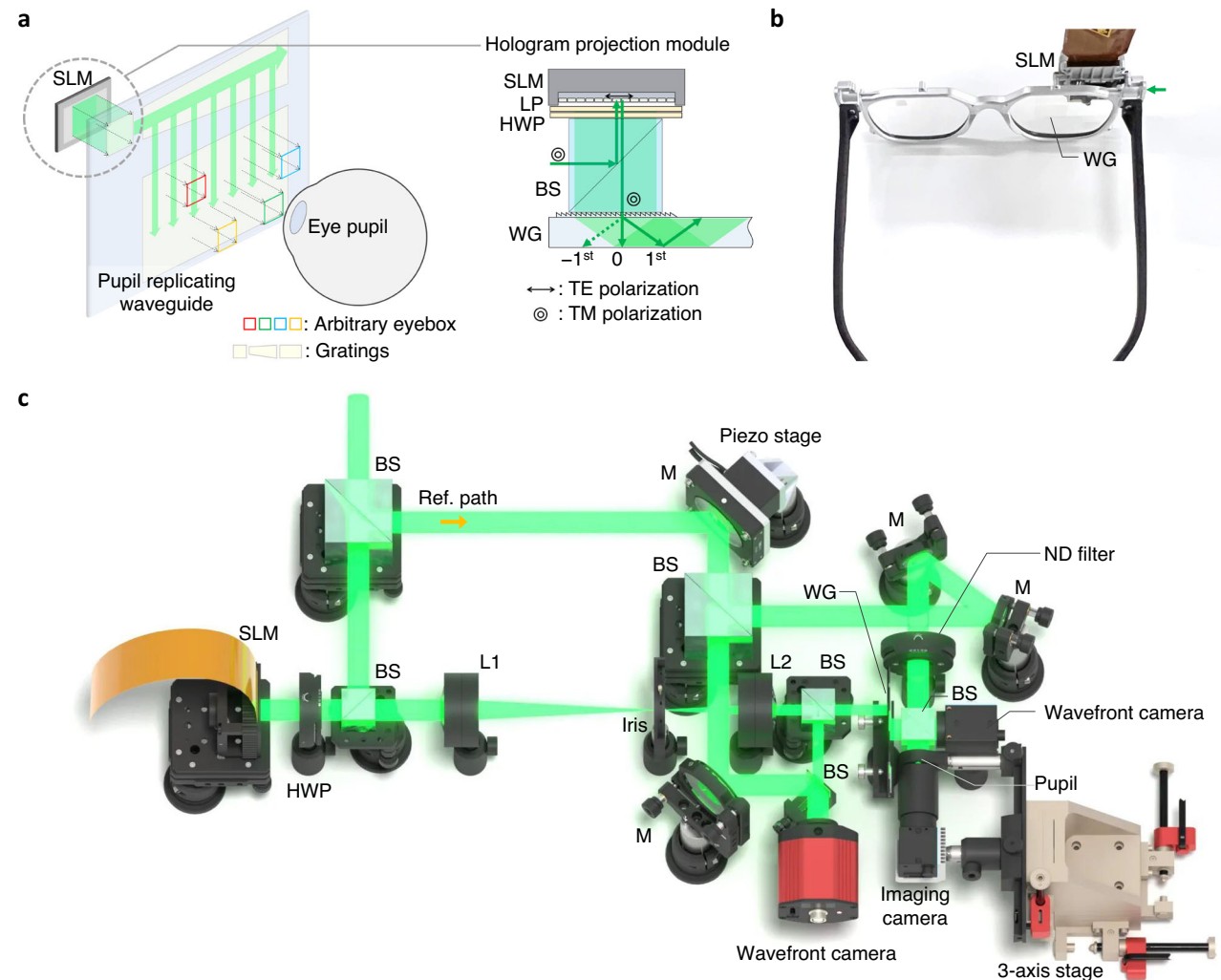

**Fig. 1 | Overview of waveguide holography. a** Conceptual illustration of the proposed display architecture using the exit-pupil expanding waveguide (WG). The hologram projection module consists of SLM, a linear polarizer (LP), a half wave plate (HWP), and beam splitter (BS) for illumination path. Apertures with different colors indicates software-steered eyebox that can be formed in the arbitrary 3D space at the out-coupler side, fully exploiting the étendue expansion capability of the waveguide. **b** A photograph of the compact prototype for the proof-of-concept. The SLM size can be further reduced since the active area is only as large as the input coupler size, which is 20% of the total SLM area. **c** A benchtop prototype built for design iteration and benchmarking the performance. L1 and L2: lenses for 4-$f$ imaging system, M: mirror. See Method for the details of the system implementation. Graphics rendered by Eric Davis.

be simplified as combination of three major optical interactions: optical propagation in the waveguide substrate; total internal reflection at the substrate boundary; and the first order diffraction at the gratings. All the three interactions are linear operators with complex-valued input and output. Also, the spatially shift invariant property can be satisfied under the assumption of homogeneous grating profiles; in other words, each grating does not have optical power or boundary. Although there are physical boundaries of the gratings, this condition can be approximately satisfied to light paths that do not encounter grating boundaries. The LSI assumption enables key advantages to model the waveguide system in terms of Fourier optics regimes. First, all the complicated interactions can be simplified as a single convolution operation. This interpretation is computationally efficient compared to tracking all the light interactions of different paths inside the waveguide, which involves manifold operations with heavy computation. Based on the assumption, we build a differentiable forward model that is useful for model calibration and CGH rendering. The analytic derivation of the convolution kernel and its gradient is provided in Supplementary Material and Supplementary Fig. 1.

Despite the advantages of the LSI assumption, typical waveguides are not perfect shift invariant systems in practice. There are a plethora of factors that alleviate the spatially shift invariant assumption, such as the non-uniformity of the grating, the surface flatness of the substrate, or the slant angle of the slab. In particular, physical boundaries of the grating introduce clipping of the wavefront and edge diffraction, resulting in different optical paths depending on the position at the input domain. In addition, defects in the grating and unwanted scattering from particles or dust all contribute to invalidating LSI approximation. Therefore, we introduce the multi-channel convolution model with complex apertures to handle the spatially variant nature of the system. Our model pipeline is illustrated in the upper part of Fig. 2, which consists of the multi-channel kernels $h$ and the complex apertures $Q$, $R$, and their visualization is presented in Fig. 3. All the apertures and kernels are complex valued 2D matrices, and their sizes are dependent on the input SLM size and output ROI size. Aperture $Q$ is intended to model the in-coupler of the waveguide, and also helps to select a different convolution path depending on the position at the in-coupler. Each $h$ is intended to emulate different possible light interaction paths inside the waveguide, which is the main source of spatial variance characteristics. Aperture $R$ additionally calibrates the intensity and phase fluctuation of the resultant field after the convolution, possibly caused by out-coupler grating or EPE grating. By merging

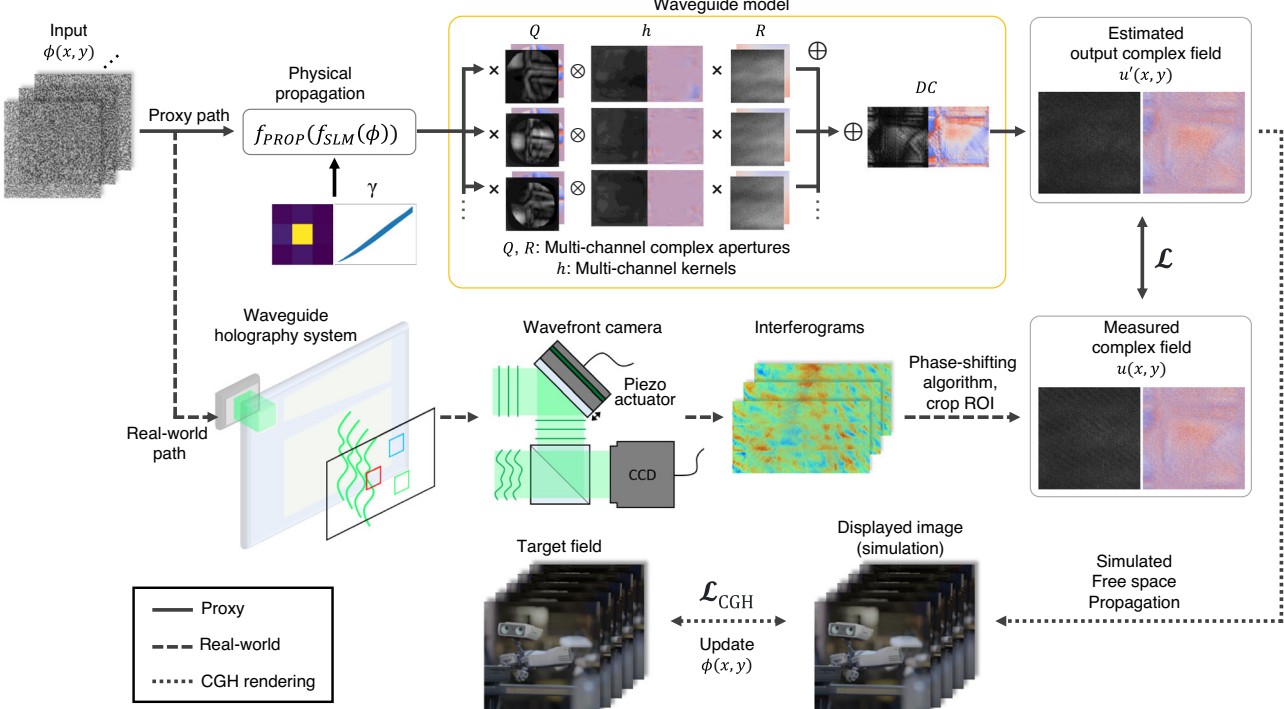

**Fig. 2 | Illustration of our modeling pipeline.** The upper part represents the proxy path that models the waveguide propagation system, while the middle part represents the real-world path that demonstrates the experimental pipeline to measure the out complex wavefront from the waveguide, which is used to generate the dataset for model calibration. Once the model is calibrated, a free space propagation can be added to render the CGH. Robot images are rendered by Tech Art team in Meta.

output wavefronts from all the paths as a linear complex-number summation, the model acquires capacity to capture spatially variant properties. The linear summation also induces a smooth transition between adjacent positions, which prevents the model from becoming too sensitive while still maintaining its differentiable property.

We also add the parameters to model SLM response and the physical propagation of wavefronts in front of the waveguide model. The SLM modeling consists of a crosstalk kernel and spatially varying phase response function. The physical propagation includes free space propagation, as well as 3D tilt and a homography changes from the alignment mismatch and aberration. Details are provided in the Method Section and Supplementary Material.

### Model calibration using complex wavefront camera

The real-world path of Fig. 2 illustrates the complex wavefront dataset acquisition pipeline. We implement the Mach-Zehnder type phase-shifting interferometer system at the out-coupler side of the waveguide that captures the interferogram of the output wavefront from the waveguide and the plane reference beam[52,53]. The complex wavefront is then retrieved using the phase-shifting algorithm. We call the interferometer system *wavefront camera* for convenience. Random phase input is used for generating the dataset since it contains all the frequency components uniformly. After the data acquisition is finished, the loss is calculated as an L1 norm between the estimated complex field and the measured complex field dataset during the training stage as:

$$L = \| u'(x,y) - u(x,y) \| . \tag{1}$$

We emphasize that the complex wavefront capture is one of the key factors that enables the precise training of waveguide propagation model. Compared with generic free space propagation, the light propagation inside the waveguide generates complicated overlapping and coherent interference of replicated wavefronts. By measuring the intensity only, it is difficult to infer the waveguide kernels and complex apertures in the model as the useful information is buried in the noisy interference pattern. With the wavefront camera, the access to phase information could successfully retrieve the coherent light interaction in the waveguide.

Also, our method offers a one-time calibration for a large 3D eyebox area. Once the model is trained, the pupil size, location, and position can be freely selected within the ROI by cropping a different area from the estimated wavefront. Additionally, eye relief of the eyebox can be changed by numerically propagating the wavefront. This is a significant difference from conventional camera-in-the-loop calibration methods, which were not practical for calibrating all the possible pupil locations and sizes separately. The size of the ROI of the model can be selected by scaling the model size to its area. Up to 7 mm square eyebox could be fitted to the model, mainly restricted by the sensor size of the wavefront camera.

### Ablation analysis

Figure 3b illustrates the estimation result of the output complex wavefront. To evaluate the contribution of the different elements consisting the model, ablation analysis is performed as presented in the left of Fig. 3c, where the ROI is set as 3.5 mm square. First, the kernel-only model consists of only a single $h$ kernel while the physical propagation module, $Q$, $R$ and $DC$ components are omitted. Then we add the physical propagation module to the kernel-only model. The single channel indicates the full pipeline including $h$, $Q$, $R$ and $DC$ components as shown in Fig. 2. We use peak signal-to-noise ratio (PSNR) and complex PSNR (c-PSNR) values to evaluate the similarity between the estimated wavefront and the measured wavefront. The complex PSNR is calculated by concatenating real and imaginary part of the complex wavefront to form a real-valued matrix. A higher value indicates that the model predicts the output wavefront with a higher

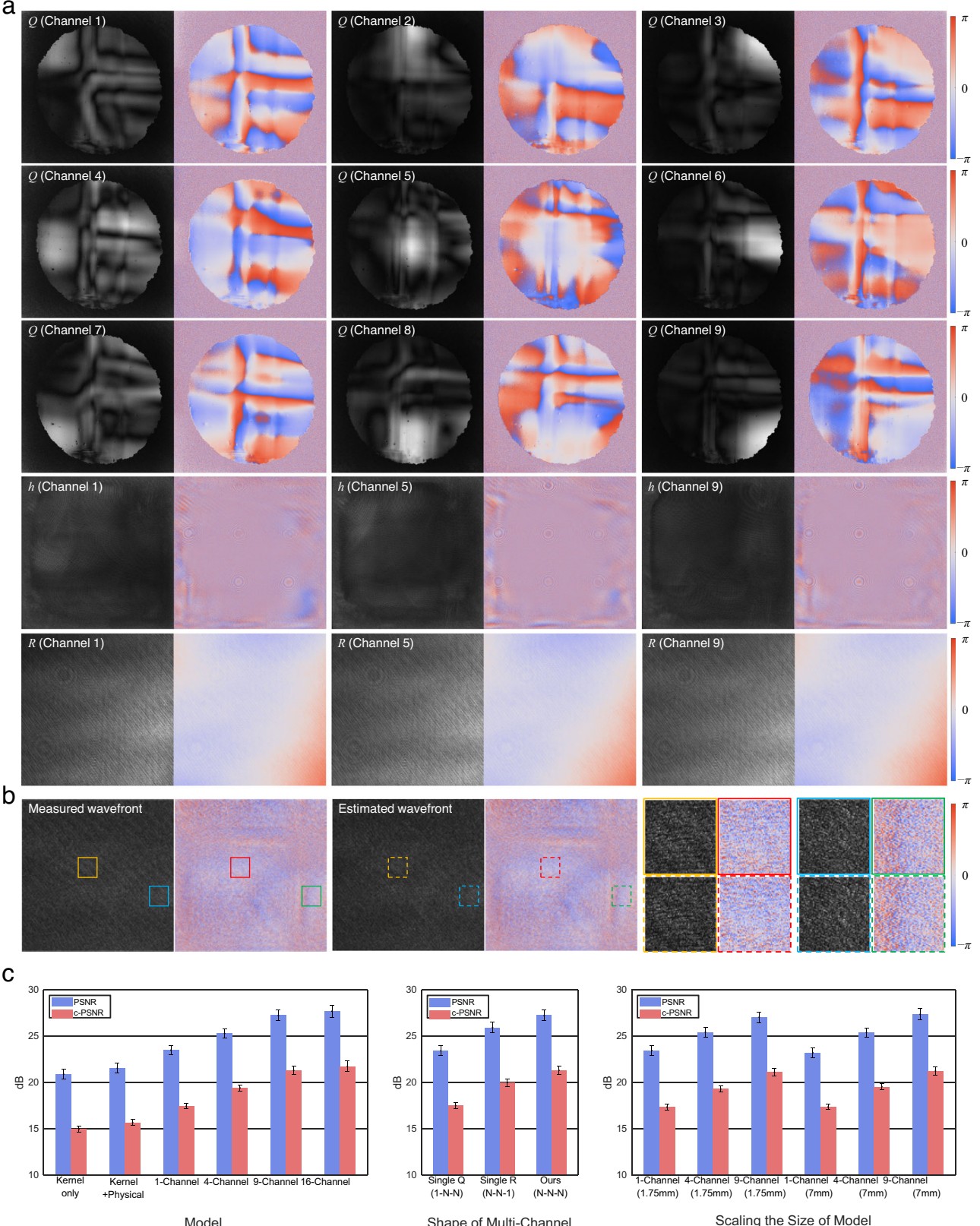

**Fig. 3 | Result of the model calibration and ablation analysis. a** Visualizations of optimized model parameters as gray-scale amplitude and color map phase images: multi-channel complex apertures $Q$ (top), multi-channel kernels $h$ (middle), and $R$ (bottom). **b** Comparison of the measured output wavefront and the estimated wavefront using the model. Orange and red insets are sampled at the same position in the eyebox domain, so as blue and green insets. Magnified images of insets are displayed in the right side for visual comparison. **c** The ablation analysis result of complex wavefront estimation performance with their standard deviation according to the variations of the modeling (left), structures of multi-channel model (middle), and scaling the size of the model by changing the ROI size and the number of channels (right).

precision. The c-PSNR tends to be lower than PSNR as only a slight change in the phase offset will result in a large error distance in the complex number domain. The result shows that the complex apertures and DC component are helpful to enhance the fidelity of the model. Also, it can be verified that multi-channel model significantly boosts the performance compared with single-channel model. The effectiveness is eventually saturated as 9-channel and 16-channel do not show a noticeable difference.

We also tested the contribution of each multi-channel parameter $Q$, $h$, and $R$. When $Q$ is set as a single channel $(1 - N - N)$, all the kernels $h$ share the same input complex aperture, therefore the model loses the capacity to handle the spatially variant property which is described above. As a result, the fidelity of prediction drops significantly as shown in Fig. 3c. Meanwhile, when $R$ channel is set as a single channel $(N - N - 1)$, the PSNR drop is relatively slight. This result aligns with the physical intuition of the modeling as $R$ is expected to capture the wavefront modulation at the out-coupler side or outside of the waveguide, which we expect to show more homogeneous response than inner waveguide interactions.

In the right of Fig. 3c, we show the scalability of the model by varying the size of the output ROI. The size of the ROI does not significantly affect the model performance which aligns with the assumptions used in the modeling. At the in-coupler side, each channel that shares a similar convolution path can be assorted spatially, as shown in the shape of $Q$. Then the pupil replication process extends the receptive field of each channel to entire out-coupler area. Therefore the required number of channels is not dependent on the out-coupler domain, but dominantly decided at the in-coupler domain.

## CGH rendering

Once the model training is finished, the CGH can be calculated by adding a numerical propagation at the end of the model pipeline with parameterized input phase of the SLM as illustrated in 2. The input phase is initialized as random and propagates through a forward path to generate the output retinal image. The loss is calculated as an L1 norm of the difference of the target image and the model output. For 3D contents, the loss can be calculated at multiple depths and added together. Focal stacks or light field images can be used to promote the accurate blurring effect or ocular parallax[24,54–56]. The loss is back propagated to update the input phase and the whole process is iterated until the estimated result reaches a certain PSNR value.

## Experimental results

Figure 4 demonstrates the experimental results captured in a benchtop prototype, where the field of view is slightly less than 11 degrees diagonally, determined by the SLM pitch size. Further system details are provided in the Method section and Supplementary Material. Capture is performed using two different methods. First, we put an imaging camera with the 3D printed entrance pupil mask with the exact size and position of the targeted ROI and capture the image directly. Second, we capture the complex wavefront in the eyebox domain using a wavefront camera and numerically propagate it to the image plane. Wavefront cameras can avoid aberration from the camera lens or alignment error because numerical propagation replaces a physical aperture and camera lens. Also, it offers precise numerical refocusing capability with a much larger depth range. However, the phase shifting process could add noise to the reconstructed image[52]. We use both capturing methods to evaluate the results. The first column of Fig. 4a is presented for comparison, where the waveguide module in the pipeline has been replaced with a generic wave propagation function that is agnostic to the waveguide. It is noteworthy that our model improves the image quality significantly even when the image is displayed at the infinity depth, where there is no explicit presence of ghost noise. When finite depth holograms are displayed, the images suffer severely from ghost noise and aberration created by

duplicated pupils without our method. The second and third column of Fig. 4a show the display results using our model captured with the imaging camera and wavefront camera, respectively, with the latter showing slightly higher resolution, albeit with a marginally diminished contrast as previously discussed. The results verify that the focus spread artifacts are solved and holograms are reconstructed at desired depths via the waveguide.

Figure 4b shows the holograms generated and captured at full depth range from zero to infinity distance. We note that Fig. 4a, b is captured at arbitrarily selected eyebox positions different from each other. Once the model is calibrated, any size and location in 3D space of the eyebox can be chosen without additional pupil calibration. We provide more results demonstrating the large eyebox and effect of pupil offset in the Supplementary Material.

3D display results are presented in Fig. 5. Display results of the compact prototype are presented in Fig. 5b, where the scene is captured through the waveguide to demonstrate the see-through quality. The 4 K SLM used in the prototype exhibits a phase flicker artifact that compromises calibration accuracy and image quality, with further details elaborated in the Method section. Figure 5c shows the temporally-multiplexed 3D results captured with the wavefront camera. CGH is rendered using focal stack target (12 planes) with accurately rendered blur and occlusion. Since our waveguide is designed for a single wavelength, we showcase pseudo-color images by merging separately captured RGB channels images in order to facilitate the intuitive visual perception of the 3D effect and image quality. View the Supplementary Movie 1[57] for the continuous focus change.

In an ideal lossless waveguide, the angular resolution of the transmitted wavefront is decided by the number of the modes and mode spacing that a waveguide can support for the monochromatic light with wavelength of $\lambda$ as:

$$\delta\theta_{res} = \frac{\lambda}{2t \tan\theta_T}, \tag{2}$$

where $t$ is the thickness of the substrate and $\theta_T$ is TIR angle field of view component. However, this does not hold in typical waveguide displays because a waveguide image combiner consists of leaky diffraction gratings with finite boundaries. Numerous beam clippings at the edges of the gratings during the pupil replication, along with clipping at the user's eye pupil, reduce the effective numerical aperture and the resolution. This beam clipping effect has been an inevitable degradation factor that sets the fundamental limit of the resolution in the waveguide display system in the most cases. Additionally, there are various non-idealities in the system that further degrade the resolution, such as aberration from the projection module or surface flatness.

We demonstrate that such resolution limitations can be overcome by adoption of holographic displays, fully utilizing the coherent nature of light. With the knowledge of light interaction in the waveguide, the phase discontinuities caused by beam clippings can be stitched to achieve smooth phase in the eyebox. Figure 6 illustrates the experimental results to display a tilted plane wave target, captured by the wavefront camera. Without using our method, severe phase discontinuities are observed in the wavefront. With the optimization, it can be visually verified that the phase discontinuity is minimized over the pupil. Also, the amplitude is optimized to be more uniform, resulting the output wavefront to be an ideal infinite conjugated plane beam. This effectively increases the numerical aperture of the display system and improve the resolution. A point spread function (PSF) and modulation transfer function (MTF) in Fig. 6 clearly visualize the improvement. The result shows sub-arc-minute resolution is achieved with over threefold increased Strehl ratio. Strehl ratio is calculated using Mahajan formula[58].

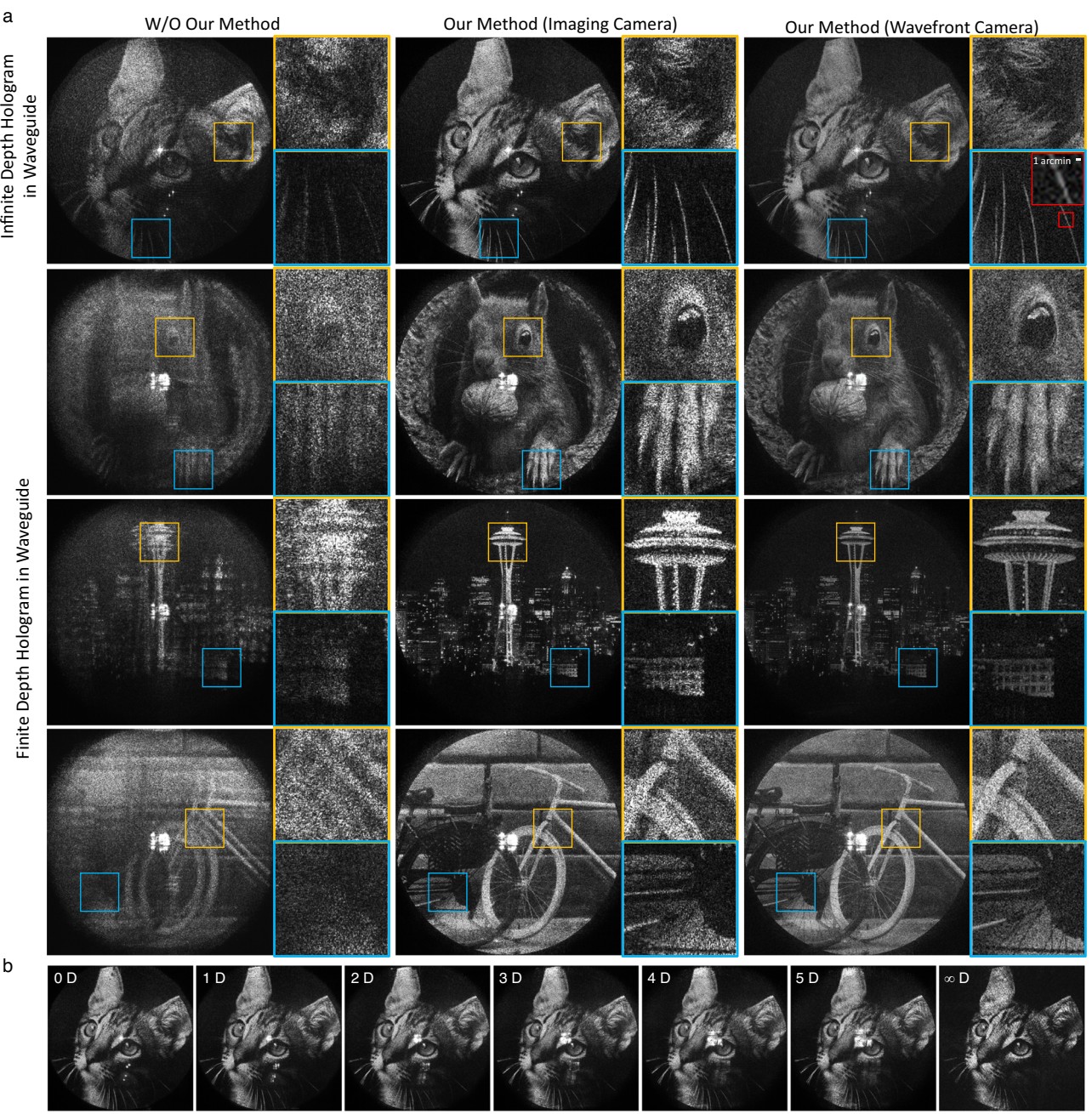

**Fig. 4 | Experimental results of waveguide holography. a** The first column illustrates the results captured without our method while the second and third columns illustrate the result with holograms generated using our model, captured with imaging camera and wavefront camera respectively. Finite depth images are displayed at 3 diopter (D) from the user's pupil. The yellow/blue insets correspond to 1.3 degree of field of view and red inset corresponds to 10 arcmin. Limited fill factor of the SLM generates a DC noise at the center of the field of view. **b** Demonstration of full depth range. The infinity diopter indicates the user's pupil plane and the image is captured directly by putting the camera sensor without lens. See Supplementary Figs. 9-14 for more results. Cat image by Lali Masriera (CC BY 2.0), Seattle skyline image by fiction-parade (CC BY-SA 2.0), bicycle image by Fiore Power (CC BY 2.0).

## Discussion

The following outlines current limitations and challenges of our work for the future research topics. In the current prototypes, the SLM causes some artifacts such as a DC noise and phase flickering. Also, the FOV is limited by the pixel pitch of the SLM. Using a projection lens could increase the FOV by sacrificing form factor; nevertheless, we choose to showcase the feasibility of the ultimate lens-free architecture, betting on future advancements in micro-display technology. With the growing expectations of AR/VR, there are ongoing efforts in academia and industry that aim to achieve breakthroughs in SLMs, such as a sub-micrometer pixel pitch[59,60], complex modulation

capability[61], and high refresh rate[62–64]. Such breakthroughs will greatly benefit the performance and scalability of the proposed architecture. We discuss related details including further miniaturization strategies and potential solutions for the DC noise in the Supplementary Material. Also, the calibration process is sensitive to mechanical perturbation by the nature of an interferometer system. Empirically, the system exhibits better robustness during the display stage than the calibration data acquisition stage. Further investigation into the system's mechanical sensitivity and improvement on the calibration algorithm would be beneficial. In the modeling perspective, a more accurate representation of the waveguide system can be studied. Our model is

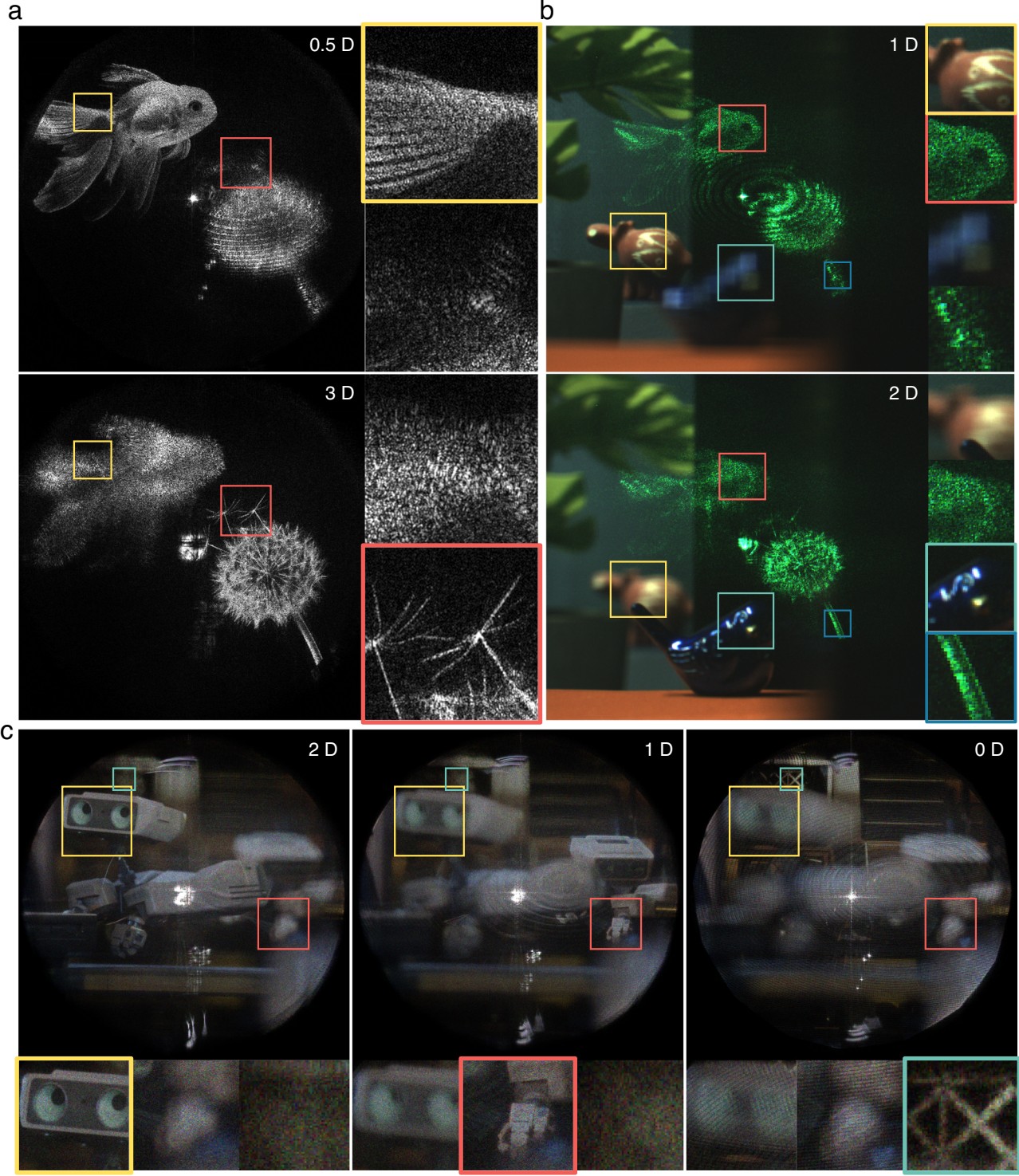

**Fig. 5 | 3D display results. a** All fish and dandelion images are displayed with a single frame of 3D hologram and captured at the different focus distances. **b** Augmented reality display results captured directly through the compact prototype glasses. The dandelion image is displayed at 3 diopter and the fish is displayed at 1 diopter. **c** Full 3D results captured with temporally multiplexed CGH (pseudo-color). RGB channel images are captured separately with the same wavelength and merged to facilitate the visual perception of the 3D effect and image quality. Each color channel has 3 sub-frames. View the Supplementary Movie 1[57] for the continuous focus change. Robot images are rendered by Tech Art team in Meta.

built on the physical intuition of the waveguide propagation process consisting of interpretable parameters. Such approach allows useful performance analysis which will be helpful for understanding the system requirements and optimizing the design. A further improvement on fidelity of modeling will lead to better calibration and display quality. On the other hand, computationally efficient modeling is another direction to be pursued. We observe some redundancies in the model parameters; for example, each kernel shares similarities in amplitude and phase shapes and the gain from increasing the number of channels saturates. Such redundancies can be reduced to shrink down the size and computation load of the model. Proposed methods can be adapted for wider applications. The multi-channel model can be

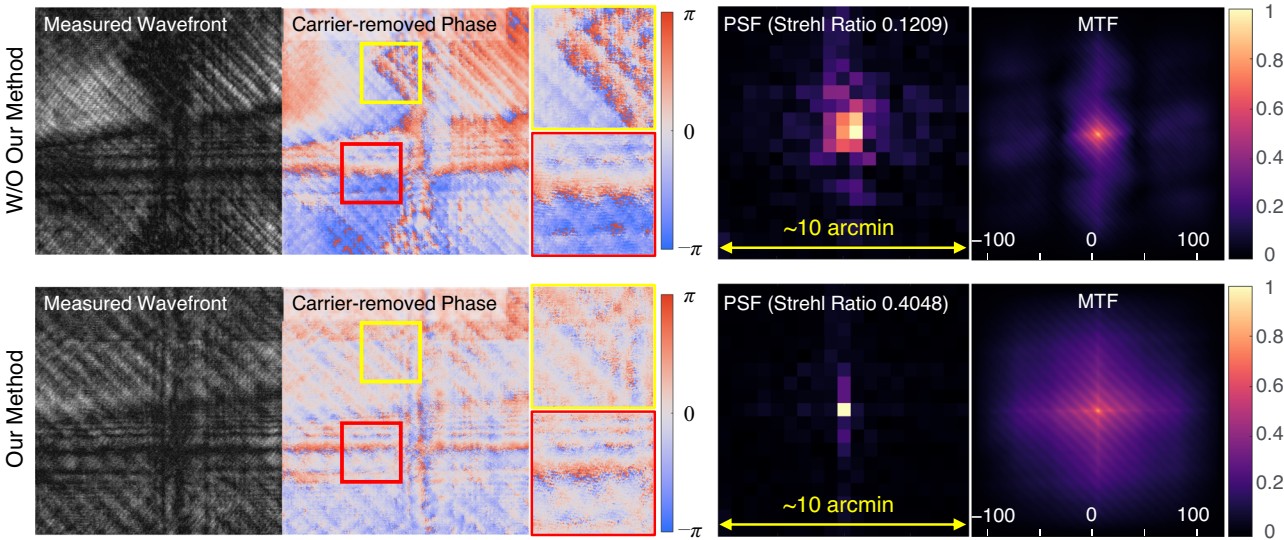

**Fig. 6 | Resolution enhancement capability of waveguide holography.** The top row and bottom row illustrate experimentally captured output wavefronts using a tilted plane wave as a target, which forms a single image point at an infinite distance, without and with our method, respectively. In the top row, the target tilted plane wave is used as the input wavefront for the waveguide, without the knowledge of the waveguide model. In the bottom row, the output wavefront is optimized to form the target tilted plane wave using our model. The carrier frequency is removed to visualize phase discontinuities by dividing with the target wavefront phase. On the right, the PSFs and MTFs are obtained from the measured wavefronts. In the PSF plots, a single pixel corresponds to 0.53 arcmin. The axis of MTF plot is in cycles per degree (cpd).

modified to capture other aspects of light interaction, such as modeling of partial coherence modes or partially polarized light. The calibration method based on complex wavefront measurements may serve as a valuable tool for calibrating other intricate holographic display systems using the coherent light source. Additionally, adoption of laser light sources for waveguide displays could potentially overcome the brightness and efficiency issues. We believe our contributions would facilitate more follow-up research to take further steps towards the ultra-compact, true 3D holographic AR glasses.

## Methods
### Waveguide fabrication
The waveguide is fabricated with a glass substrate with refractive index of 1.5 and 1.15 mm thickness. The thickness of the substrate is selected based on the simulation to achieve a good performance and still retain the thin form factor. The center TIR angle is set at 50 degrees with the center wavelength of 532 nm. The waveguide is designed to support 28 degrees of diagonal field of view with an outcoupler size of 16 × 12 mm. The waveguide samples are fabricated using a nano-imprinting method which is suitable for mass production. The specifications of surface relief gratings such as shape, slant angle, and aspect ratio are fine-tuned using rigorous simulations to achieve spatial and angular uniformity at the eyebox domain (see Supplementary Fig. 3). In general, targeting higher uniformity reduces the grating efficiency and thus trades overall efficiency. We set the merit function to balance between uniformity and efficiency, to achieve over 5% of end-to-end throughput efficiency on average and maximize the uniformity. The grating structure is designed as a saw-tooth shape to minimize the unwanted diffraction orders[65,66], however it diffracts some portion of light as −1st order. Therefore, we used a beam splitter in the hologram projection module and the light source was expanded outside of the system. Further details are presented in Supplementary Material.

### Implementation of prototypes
In the optical benchtop prototype, a 532 nm Cobolt Samba 1500 mW laser is used as a light source, a Piezosystem Jena PZ-38 as a piezo actuator for phase shifting digital holography, and a Meadowlark E-series 1920 × 1200 SLM. In de-magnifying relay system, 150 mm (L1) and 75 mm (L2) focal length lenses are used. 62.5% of the SLM area (1200 × 1200 pixels) is used to generate input wavefronts. We built two wavefront cameras in the system; one for capturing the waveguide output wavefront, and the other for capturing relayed SLM to optimize SLM parameters and homography in the model. Both wavefront cameras share the same piezo actuator for phase shifting. A neutral density (ND) filter is used to attenuate the light intensity in the reference path for both wavefront cameras. Details of calibration algorithm is presented in Supplementary Material (see Supplementary Fig. 2). To capture the result images, a 3D printed pupil aperture (3.4 mm square) and an imaging camera are mounted on 3-axis motorized stages and placed at the copy of the output wavefront, duplicated using a beam splitter. The benchtop prototype has 11 degrees of diagonal field of view with 16 × 12 mm eyebox size. The photograph of the system is presented in Supplementary Fig. 7.

In the compact prototype, a 4-*f* relay system is eliminated and instead we used a 4 K SLM with 3840 × 2160 resolution and 3.74 μm pixel pitch, supporting 12 degrees of field of view. Only 20% of the SLM area (1300 × 1300 pixels) is actively used while other pixels are deactivated. The image quality degradation in the compact prototype is majorly caused by the SLM performance. The SLM has about 10% of phase flicker, which severely degrades the fidelity of model calibration compared with the benchtop prototype since the complex wavefront calibration method is highly sensitive to the phase error. A poorly calibrated *DC* component causes interference pattern artifacts at the far distance. Also, the SLM has more severe fringe field effects that further deteriorate the image quality. We identified that unfiltered high-order diffraction from the SLM is not a major cause of the quality degradation as they are above the sampling rate for wavefront capturing. The compact prototype experiment is performed on the optical table and a collimated laser is provided externally through a 5 mm beam splitter to the hologram projection module. For calibration, another beam splitter is placed at the eyebox of the waveguide to combine the reference beam.

### Details of the algorithm
The physical propagation module in Fig. 2 consists of a crosstalk kernel of the adjacent SLM pixels, spatially varying phase modulation

function of the SLM ($\gamma$), free space propagation with 3D tilt[67,68], and a homography transformation function. The crosstalk kernel has $3 \times 3$ size and $\gamma$ is modeled as a polynomial with 18 coefficients. The free space propagation function has 3 parameters including 2D tilt angles and the propagation distance. The homography transformation function is up to second order with 12 coefficients. The physical propagation module is calibrated in advance in the benchtop prototype using the wavefront camera placed at the relayed SLM and then used as the initial value of the model calibration to accelerate the calibration. In the waveguide model, the size of $Q$ is selected to cover the physical size of the in-coupler grating, which is $1200 \times 1200$. The size of $R$ and $DC$ is selected to be equal to the ROI size. The side length of Kernel $h$ is set as summation of $Q$ and $R$. For the model calibration, about a thousand captured wavefronts are used as a dataset. In the CGH rendering stage, the loss starts to converge around 1000 iterations and we run up to 3000 iterations. Further implementation details are provided in Supplementary Material.

## Data availability
The source data used for Figs. 3 and 6 have been deposited in Figshare under accession code https://doi.org/10.6084/m9.figshare.22672519[57].

## Code availability
The code used for modeling the hologram propagation in waveguides will be made publicly available (on GitHub) along with the paper. Additional codes are available from the corresponding authors upon request.

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

## Acknowledgements

Giuseppe Calafiore, Heeyoon Lee, and Alexander Koshelev have contributed to the design and fabrication of the waveguide used in this work. Clinton Smith designed the compact prototype and provided engineering support. We also thank the valuable discussion provided by Grace Kuo and Suyeon Choi. Julia Majors provided proofreading and valuable feedback. Eric Davis rendered the graphics in Fig. 1c. In Fig. 4, the cute cat image by Lali Masriera (CC BY 2.0), the beautiful Seattle skyline image by fiction-parade (CC BY-SA 2.0), and the nice bicycle image by Fiore Power (CC BY 2.0). 3D contents in Fig. 5c are rendered by Tech Art team in Meta. The research is supported by Meta.

## Author contributions

C.J. and K.B. initiated the project, designed the architecture, conceived algorithms, and conducted the experiments. M.C. conducted the experiments. D.L. and B.L. advised and supervised the project. C.J. wrote the initial draft of the manuscript and all authors reviewed the manuscript.

## Competing interests

C.J. and D.L. are currently employees of Meta. K.B. was an employee of Meta. We declare that M.C. and B.L. do not have any conflicts of interest.
