## [Peer Review File · Nature Communications]

Waveguide holography for 3D augmented reality glassesReviewer #1 (Remarks to the Author):

This paper proposes a new approach to 3D displays which the authors call waveguide holography. 3D display technology is currently an area of huge interest and the work described here certainly contributes the field. I have a fundamental question about the paper - which is about its novelty. The concept of 3D displays based on holography is not new, nor are 3D displays based on waveguides. That's fine - the authors make this clear. But the paper is fundamentally about bringing the two together and the authors cite very few papers on the topic of both of them together - whereas a quick search indicates there has been quite a lot of research in this area. My concern increased when I read this sentence "In this study, we present a novel lens-less near-eye display system dubbed waveguide holography that" which I take to mean they are proposing the term waveguide holography. Again a quick search indicates that this is not a new term.

So - I think fundamentally the authors need to do more work on clarifying what is new here, and also explaining the fundamental ideas. As I read the current version - to me it is a detailed technical paper (which is fine) which should go to a more specialist optics journal. The authors might disagree with this - and I'd be happy to see the paper again if they can address my fundamental point about novelty and their claim that this is a new concept - rather than a refinement of an existing technique.

Final point. The authors might like to proof read their text. There are many very minor grammatical errors.

Reviewer #2 (Remarks to the Author):

This manuscript investigates a timely topic of computational holographic displays, whose research field has also become highly crowded recently in both the optics and the graphics communities. The core of this work lies in exploring the combination of a waveguide combiner with computational tricks to enable AR experience with the conventional holographic display engine. The motivation is well-received. As a more algorithmic-driven innovation, the key contribution should be investigating the modeling of the wavefront (and its propagation) with the pupil-replicating waveguide. To achieve this, the wavefront sensing-aided calibration is presented and a number of 2D and a set of 3D (2.5D) results are presented. No major technical flaw is found. With respect to the hardware side, the work is built upon mostly off-the-shelf devices or solutions. The implementation of such two prototypes is definitely non-trivial, which I really appreciate the authors' effort. While overall, the manuscript is in a reasonably good shape in terms of concept, writing, technical flow, and exposition, there are several comments/concerns that the authors should clarify or elaborate so as to lead to a more convincing scientific article.

1. To my best knowledge (with experimental observation and insights from several industrial players in the optics combiner field), waveguides, no matter volume hologram type or surface-relief grating type, would show noticeable impact on the polarization status of the input wavefront. This effect is often a bit complicated and hard to be disentangled since many parts/components could contribute to it. That said, the wavefront coming out from the waveguide and continuing to propagate to the wavefront cameras is no longer accurately sharing the same polarization as the reference beam. I believe adding a polarizer right after the waveguide out-coupler (in front of the cameras) would be an easy but crucial way to lead to a much higher SNR wavefront calibration. I am curious why the authors didn't do this. Even the authors claimed in the supplement that the polarization was ignored reasonably, it would be necessary to present a comparison with/without this extra polarizer.
2. Figure 2 seems very insightful. I would be curious why the authors would set up so many different Q and R kernels? Frankly, Q represents the modulation that happens on the in-coupler end? If that's the case, shouldn't it be the same for all channels which seems to be more physics-inspired and saving computing memory/resource? The same concern applies to the out-coupler end. Have the authors tried a different setting of this multi-channel configuration, like 1-N-1 or 1-N-N? It would be really helpful to see a more solid justification on the multi-channel convolution model representation, since this is really the most interesting contribution.

3. The 3D results (Figure 6 and 7) unfortunately suffer from the poor quality, in terms of both resolution and defocus blur effect. Although I fully understand this could be very challenging, elaborated discussion could aid to the argument. Yet, as holographic displays are essentially 3D-driven, more 3D results should be included in the paper. Now the entire manuscript with over 16 pages has only one set of results regarding 3D, or more rigorously speaking, 2.5D.
4. Following the above comment, considering the current image quality and the lack of natural 3D behavior, the title sounds a bit over-claimed and too general. First of all, "waveguide holography" isn't an entire novel concept. Many prior work, including several cited papers, for example [Yeom 2015; Jan 2018; Choi 2021], has already discussed the combination of waveguides and holographic display engines. In addition, the second part of the title "Towards True 3D Holographic Glasses" holds for almost every research paper that was recently published or is to be published in this field. As such, for a scientific article (not a PR story-telling article) on such a high-impact journal, I would assume a more specific, appropriate title should be made. In the meanwhile, I would suggest rewording the tone accordingly across the introduction and abstract to sound more humble and solid.
5. The authors claim to retain the resolution advantage of holographic displays. However, there is no results of standard resolution charts or patterns being presented. I would suggest adding the comparison with/without the proposed model/calibration using a resolution chart target. This would be insightful in convincingly showing not only resolution but contrast performance. Both of these would be highly significant for AR glasses.
6. Regarding the super-bright spot in the center of almost all results, is it really just the DC term issue of the SLM? Was Fresnel holography or Fourier holography being used? Or is there any practical trick to partially mitigate it in the generation of holograms? I believe this issue matters significantly for near-eye displays. Please clarify in the manuscript.
7. The termed "waveguide design" part seems a bit brute-forced. Basically, the authors still rely on existing grating equation and cherry-pick a thickness? Since the authors were fabricating their own customized waveguide anyways, it would be great to note the discussion of optimizing a waveguide in a more deterministic manner, including but not limited to thickness, refractive index, grating parameters, etc. Please clarify.
8. How well would the wavefront calibration and the multi-channel model be generalized to different types of diffractive waveguides?
9. Do the authors have an idea if it could actually be fine to remove the high-cost and maybe ambiguous wavefront camera settings here? Instead, how good the results would be if just using conventional intensity sensors to supervise the model learning?
10. It would be beneficial to have the actual photographs and layouts of the two prototypes in the suppl. at least, rather than just schematic diagrams.
11. I appreciate the detailed elaboration of future possibilities in Figure 10, although very interesting, this would not be justified as the core contribution of current manuscript. As such, maybe shorten these paragraphs a bit in the main text and yield the detailed elaboration to the supplement?
12. I am also curious how would recent emergence of AI/DNN fit into this work, considering such an amazing amount of AI-driven holographic display solutions have been recently launched in academia?

=====

Author's Response of Paper Titled:

Waveguide Holography: Hologram Propagation via Pupil-Replicating Waveguides

=====

Submission ID: NCOMMS-22-37509

Original title: "Waveguide Holography: Towards True 3D Holographic Glasses"

Original submission date: September 30th, 2022

Review received on: February 9th, 2022

Revision submitted on: April 21st, 2023

Reviewer #1 (Remarks to the Author):

This paper proposes a new approach to 3D displays which the authors call waveguide holography. 3D display technology is currently an area of huge interest and the work described here certainly contributes the field. I have a fundamental question about the paper - which is about its novelty. The concept of 3D displays based on holography is not new, nor are 3D displays based on wave-guides. That's fine - the authors make this clear. But the paper is fundamentally about bringing the two together and the authors cite very few papers on the topic of both of them together - whereas a quick search indicates there has been quite a lot of research in this area. My concern increased when I read this sentence "In this study, we present a novel lens-less near-eye display system dubbed waveguide holography that" which I take to mean they are proposing the term waveguide holography. Again a quick search indicates that this is not a new term.

So - I think fundamentally the authors need to do more work on clarifying what is new here, and also explaining the fundamental ideas. As I read the current version - to me it is a detailed technical paper (which is fine) which should go to a more specialist optics journal. The authors might disagree with this - and I'd be happy to see the paper again if they can address my fundamental point about novelty and their claim that this is a new concept - rather than a refinement of an existing technique.

Final point. The authors might like to proofread their text. There are many very minor grammatical errors.

===== Author's Response to Reviewer #1 =====

We appreciate reviewer's time and effort reviewing our manuscript. We understand that our manuscript may not have effectively conveyed the novelty of our approach, and we would like to take this opportunity to further clarify our contributions and problem statement. Let us begin with summarizing the problem statement and our contributions/novelty that we have provided in the previous manuscript as follows.

1. Problem statement

In the **Introduction Section**, we discuss the limitations of two promising technologies; waveguide display, which is a dominant technology for augmented reality glasses, and holographic display, which is believed to be the ultimate 3D displays.

Waveguide displays:

[Line 35] "...waveguide display can only display a single depth, normally infinity conjugate image. If finite-conjugate image is projected to the waveguide, the pupil replication process produces copies of different optical path and aberration that create severe ghost noise. ... Dual/multi-imaging planes waveguide architecture have been studied, but they come with the cost of

degraded performance, bulkier form factor and additional hardware constraints”

[Line 31] “...it is challenging to achieve enough brightness using conventional light sources such as micro light-emitting diode (LED)... Laser light source can be much more power efficient than micro LED, however, the coherent light interaction causes several artifacts”

Holographic displays:

[Line 47] “...However, the architecture of near-eye holographic display for augmented reality remains as unsolved problem because of the étendue limitation, and it is even more difficult to achieve a glasses form factor.”

[Line 51] “...Especially, practical ergonomic design does not allow enough bandwidth of oblique projection angle and space in the temple side, which makes the glasses form factor even more stretched goal.”

Especially, we would like to emphasize again that vergence-accommodation conflict is a well known problem for AR/VR research field (Erkelens, I. M. & MacKenzie, K. J. 19-2: Vergence-accommodation conflicts in augmented reality: Impacts on perceived image quality. SID Symp. Dig. Tech. Pap. 51, 265–268, 2020), yet the solution for full 3D waveguide displays has not been introduced despite lots of investment in the industry. Kress et al. describes providing accommodation using waveguide displays is **“quasi-impossible”** in the review article (Kress, Bernard C., and Ishan Chatterjee. "Waveguide combiners for mixed reality headsets: a nanophotonics design perspective." Nanophotonics 10, no. 1 (2021): 41-74.) in **Section 5.2.5 Focus spread in waveguide combiners:**

“...It is quasi-impossible to compensate for such focus shift over the exit pupils because of both spectral spread and field spread over the exit pupils, as discussed previously...”

2. Novelty

We emphasize that we are the first to demonstrate the full depth range solution for pupil expanding waveguide, as presented in **Fig. 6** (previous version) and we note it in **Introduction** and **Experimental Results** sections:

In **Introduction**,

[Line 61] “...Our approach is fundamentally different from previous works in that pupil expansion is enabled to provide large eyebox with about a millimeter thick waveguide. The core idea of waveguide holography is to model the coherent light interaction inside the pupil replicating waveguide. We introduce a novel multi-channel kernel modeling of waveguide wave propagation. The precise model calibration is enabled by complex wavefront capturing system and algorithm based on phase-shifting digital holography. As a result, the combination of the two state-of-art display technologies; waveguide image combiner and holographic displays, allows compact glasses like form factor, as well as displaying true 3D images. We demonstrate that out-coupled wavefront from the waveguide can be precisely controlled by modulating the input wavefront using our model. Our system is experimentally verified to support a full depth range with per-pixel depth reconstruction, diffraction limited resolution, as well as étendue expansion which enables a large 3D eyebox with a full native field of view of spatial light modulator with software-steered exit-pupil. We also present a detailed analysis and intuitive

discussion for the architecture design and scalability. We conclude the study with a discussion on some limitations as well as interesting future works”

In Results,

[Line 186] “We demonstrate that our prototype can generate image at arbitrary depth from zero to infinity distance from the eyebox plane, confirming the full 3D image can be displayed in the waveguide only modulating the SLM.”

We emphasize the result of etendue expansion (large eyebox) and 3D eyebox one-time calibration presented:

In Results,

[Line 9] “...the system can fully utilize the expanded étendue and achieve a software-steered eyebox without mechanical steering as large as conventional waveguide image combiners can provide.”

[Line 137] “...Once the model is trained, pupil size, location, and position can be freely selected within the ROI by cropping different area from the estimated wavefront. In addition, eye-relief of the eyebox can be changed by numerically propagating the wavefront. This is a major difference from conventional camera-in-the-loop calibration methods which were impractical to calibrate all the possible pupil locations and size independently.”

Additionally, we demonstrate the resolution enhancement:

In Results,

[Line 205] “...we demonstrate that our method can overcome such resolution limit. With the knowledge of light interaction in the waveguide, the phase discontinuities caused by beam clippings can be stitched to achieve smooth phase in the eyebox. ... As a result, sub-arc-minute resolution is achieved with over threefold increased Strehl ratio.”

We also summarized the above contributions in **Abstract** and **Discussion** sections:

In Abstract,

“...The proposed system can display true 3D holographic images through see-through pupil-replicating waveguide combiner as well as providing a large eye-box.”

In Discussion,

*[Line 237] “...we propose a novel AR near-eye display architecture by combining two state-of-art display technologies; waveguide display, which is the industry norm technology aiming augmented reality glasses, and holographic display, which is believed to be the ultimate 3D display technology. We have demonstrated to display **holographic images at arbitrary depths and eyebox position through pupil replicating waveguide for the first time**, opening the new possibility towards the true 3D holographic glasses.”*

As shown above, we believe we have emphasized our fundamental idea and novelty distinguished from previous research. However, we realize that our presentation method might not have facilitated efficient comprehension. In the revised version, we will enhance the paper's organization and presentation to emphasize its novelty more clearly.

From here, we would like to address reviewer's main concern on novelty and citations as reviewer mentions:

"I have a fundamental question about the paper - which is about its novelty. ... But the paper is fundamentally about bringing the two together and the authors cite very few papers on the topic of both of them together - whereas a quick search indicates there has been quite a lot of research in this area. ... Again a quick search indicates that this (waveguide holography) is not a new term. "

Specifically, the reviewer points out that there are lots of search results containing similar keywords. Indeed, "waveguide, guided-wave, waveguide display" and "holography/holographic/hologram" are two emerging topics, and when searched together, they produce a lot of results. However, we believe that a significant number of these search results are not actually relevant to our work, but merely have similarities in titles or terms. Since the reviewer does not provide specific references, we have meticulously examined previous studies in the search results using various combinations of keywords and categorized them to discuss the differences as follows:

- a) **Holographic waveguide.** This is the most frequently found category from our search. This refers to pupil replicating waveguide image combiners made of holographic material such as volume Bragg gratings (VBG) or polarization volume hologram (PVH), instead of surface relief grating (SRG). We do not find this research relevant to our work, as the word "holographic" **does not refer holographic display** (which is a technology that actively modulates the wavefront to produce virtual 3D image at arbitrary depth), but it only refers the holographic material that is able to records the light interference pattern.

e.g.)

R. Fernández, S. Bleda, S. Gallego, C. Neipp, A. Márquez, Y. Tomita, I. Pascual, and A. Beléndez, "Holographic waveguides in photopolymers," Opt. Express 27, 827-840 (2019)

Wang, Jian Gang, Zhan Jun Yan, and Wen Qiang Li. "Optical Design of Waveguide Holographic Binocular Display for Machine Vision." Applied Mechanics and Materials. Trans Tech Publications, Ltd., September 2013.

- b) **Waveguide Holograms.** The term has been used since the early 1970s, and **it refers to the optical element or integration method of diffractive gratings** (in-coupler or out-coupler) on the waveguide. The research focuses on in-coupling or out-coupling light into the waveguide using the hologram (or grating in general). In modern waveguide display industries and academia, this terminology is not commonly used anymore.

Some studies have demonstrated the use of DOE with computer-generated hologram patterns at the out-coupler grating to project images at a far distance. However, this is fundamentally different from our work for the following reasons: first, this is only available to project a **static** image, thus it cannot be used as a holographic display. Second, **the wavefront is modulated at the out-coupler side of the waveguide**; in other words, the waveguide is used for illumination purposes only, and the hologram is agnostic to the modeling of waveguide propagation.

Meanwhile, one of our main contributions is to model the complicated light interaction and propagation in pupil-replicating waveguides.

e.g.)

Toshiaki Suhara, Hiroshi Nishihara, Jiro Koyama, Waveguide holograms: A new approach to hologram integration, *Optics Communications*, Volume 19, Issue 3 (1976)

Johan Backlund, Jörgen Bengtsson, Carl-Fredrik Carlström, and Anders Larsson, "Incoupling waveguide holograms for simultaneous focusing into multiple arbitrary positions," *Appl. Opt.* 38, 5738-5746 (1999)

A. Wüthrich and W. Lukosz, "Holography with guided optical waves," *Appl. Phys.* 21, 55–64 (1980)

- c) **Waveguide holography.** We found two papers using this term, both written by the same first author, that essentially describe similar research to the category b). Although the authors claim that their research could potentially be extended to AR displays in the future by using an SLM instead of a static DOE, it has not been proven to be possible to our knowledge, and we believe this is fundamentally different from our work for reasons similar to described in b). First, the waveguide is used as an illumination source, and no information is carried via the waveguide until the light is outcoupled. Second, the SLM cannot simply be placed at the out-coupler side for AR near-eye displays because it would obstruct the real-world view. Additionally, we believe the term is not widely used or recognized enough to prevent it from being used as the title of our paper. However, we agree that it is fair to acknowledge these previous works and clarify the differences in our revised manuscript.

e.g.)

Zhiqin Huang, Daniel L. Marks, and David R. Smith, "Out-of-plane computer-generated multicolor waveguide holography," *Optica* 6, 119-124 (2019)

Zhiqin Huang, Daniel L. Marks, and David R. Smith, "Polarization-selective waveguide holography in the visible spectrum," *Opt. Express* 27, 35631-35645 (2019)

- d) **Guided-wave holography.** This term is found in a paper published in 1992, which is about grating design for beam splitting application.

e.g.)

Jyrki Saarinen, Juhani Huttunen, Antti Vasara, and Jari Turunen, "Computer-generated guided-wave holography: application to beam splitting," *Opt. Lett.* 17, 300-302 (1992)

- e) **Holographic displays with lightguide slab (non-pupil replicating waveguides).**

This category is the most relevant research that we could find, even though the titles and terminologies are sometimes dissimilar. We have cited several papers [41-44], as these papers share a similar motivation to transmit holograms to the user's eye via waveguide. However, there are several important differences in the methods, as we have clearly stated in the manuscript. First and foremost, previous research uses non-pupil-replicating waveguides. In such cases, the architecture is similar to bird-bath type combiners with a diffractive optical element because there is only a single optical path and exit pupil that delivers the image to the user's eye. To ensure this condition, the overlap between wavefronts should be avoided in the waveguide propagation, which imposes some fundamental limitations on scalability: the eyebox

position is fixed and small, and the field of view is also limited. Kress and Chatterjee also discuss this in their review article (Kress, Bernard C., and Ishan Chatterjee. "Waveguide combiners for mixed reality headsets: a nanophotonics design perspective." *Nanophotonics* 10, no. 1 (2021): 41-74) as:

"...The image over the input pupil can, however, be located in the near field when no pupil replication scheme is performed in the guide, ... (yielding a small FOV and small EB)"

Additionally, such designs require very thick slabs (e.g., 3 – 8 mm) compared to pupil-replicating waveguides (typically ~1 mm). We would like to emphasize again that thickness is one of the most important aspects characterizing the form factor of AR glasses. Often, these thick slabs are referred to as "lightguides" to distinguish them from pupil-replicating waveguides. The compact form factor is one of our main contributions, as well as the etendue/eyebox expansion capability of thin pupil-replicating waveguides.

We have cited 4 relevant studies and stated the differences in **Holographic displays using waveguide** subsection:

[Line 56] "However, there has been a fundamental limitation on scalability because the method is not intended to support pupil replication. The light guiding slab needs to be thick enough to avoid the replication, or the overlapped wavefront is scrambled, creating severe artifacts such as multiple ghost images and low contrast. Consequently, eyebox and field of view are limited to be small in such architectures."

[Line 61] "Our approach is fundamentally different from previous works in that pupil expansion is enabled to provide large eyebox with about a millimeter thick waveguide. The core idea of waveguide holography is to model the coherent light interaction inside the pupil replicating waveguide..."

- f) **Pseudo hologram and waveguide displays.** Sometimes, translucent AR contents displayed using waveguide combiner are described as "holograms," similar to Pepper's ghost (or pseudo hologram).

Among the above categories, we find it difficult to place our work within any subgroup. Although, we recognize that we could better highlight our novelty and acknowledge more previous research by clearly distinguishing our work from them. In the revised version, we have further clarified the difference from the category e) in the **Introduction**, as well as acknowledge their contributions. Additionally, we have included more references from categories b) and c), along with clarifications of the differences. The category a) was already cited in the previous version, but we have added more references in the **Introduction**. We hope our revisions will improve the presentation and convey the novelty more clearly and accurately.

Revisions made:

Abstract:

We introduce a new near-eye display concept that incorporates a waveguide combiner, a spatial light modulator, and a laser light source. We demonstrate controlling the output wavefront of a waveguide by modeling the coherent light interactions inside the waveguide combiner and modulating the input wavefront using a spatial light modulator. The proposed method enables the display of full depth range 3D holographic images via a pupil-replicating waveguide with a large software-steerable eyebox. The method also offers additional advantages such as compact form factor realized with a lens-less holographic projector, and resolution enhancement capability by suppressing phase discontinuities from pupil replication. We build prototypes to verify the concept with experimental results and conclude the paper with summary and discussion. The combination of the two state-of-art display technologies; waveguide image combiner and holographic displays, unlocks new potentials towards true 3D holographic augmented reality glasses.

In Introduction – Waveguide Image combiner,

“...Despite the unique advantages of waveguide image combiners, there are some limitations to be addressed. First, waveguides can only convey a fixed depth, typically as infinity conjugate images. If finite-conjugate images are projected into the waveguide, the pupil replication process produces copies of different optical paths and aberrations that create severe ghost noise, which is often called focus spread effect. Generating natural focus cues and addressing the vergence-accommodation conflict are among the challenging goals of AR in the pursuit of realistic and comfortable visual experiences. Dual or multi-imaging plane waveguide architectures have been studied, but inherently lead to a bulkier form factor and diminished performance, along with added hardware restrictions...”

In Introduction – Holographic displays using waveguide,

“There have been early efforts to use waveguides as an illumination source to produce a projection pattern or image formed by an out-coupler grating with an embedded hologram pattern. Because only a static image could be displayed and no information was carried inside the waveguide until out-coupled, this approach was not suitable for augmented reality display purposes, but represented a very early stage attempt to combine waveguides and holograms together.

Recently, researchers have attempted to implement dynamic holographic displays using the light guiding slabs, with further efforts being made to compensate for aberrations and improve image quality. While they share similar motivations of transmitting holograms via waveguides, there are fundamental limitations on scalability because the method is not intended to support pupil replication; in other words, the focus spread effect remains unsolved. The light guiding slab must be thick enough to avoid replication or the overlapped wavefront becomes scrambled, creating severe artifacts such as multiple ghost images and low contrast. As a result, thick substrates (3-8 mm) are chosen, which would not be suitable for a glasses form factor. Additionally, the eyebox and field of view are fundamentally limited to be small in such architectures...”

Newly added references

Zhiqin Huang, Daniel L. Marks, and David R. Smith, "Out-of-plane computer-generated multicolor waveguide holography," *Optica* 6, 119-124 (2019)

Toshiaki Suhara, Hiroshi Nishihara, Jiro Koyama, "Waveguide holograms: A new approach to hologram integration," *Optics Communications*, Volume 19, Issue 3 (1976)

Johan Backlund, Jörgen Bengtsson, Carl-Fredrik Carlström, and Anders Larsson, "Incoupling waveguide holograms for simultaneous focusing into multiple arbitrary positions," *Appl. Opt.* 38, 5738-5746 (1999)

R. Fernández, S. Bleda, S. Gallego, C. Neipp, A. Márquez, Y. Tomita, I. Pascual, and A. Beléndez, "Holographic waveguides in photopolymers," *Opt. Express* 27, 827-840 (2019)

Erkelens, I. M. & MacKenzie, K. J. 19-2: Vergence-accommodation conflicts in augmented reality: Impacts on perceived image quality. *SID Symp. Dig. Tech. Pap.* 51, 265–268, DOI: <https://doi.org/10.1002/sdtp.13855> (2020)

Furthermore, we have made numerous updates to the overall manuscript and Supplementary Material, which we believe further clarified the novelty and contributions. For more details on the revisions made, please refer to our response to Reviewer #2.

Reviewer #2 (Remarks to the Author):

This manuscript investigates a timely topic of computational holographic displays, whose research field has also become highly crowded recently in both the optics and the graphics communities. The core of this work lies in exploring the combination of a waveguide combiner with computational tricks to enable AR experience with the conventional holographic display engine. The motivation is well-received. As a more algorithmic-driven innovation, the key contribution should be investigating the modeling of the wavefront (and its propagation) with the pupil-replicating waveguide. To achieve this, the wavefront sensing-aided calibration is presented and a number of 2D and a set of 3D (2.5D) results are presented. No major technical flaw is found. With respect to the hardware side, the work is built upon mostly off-the-shelf devices or solutions. The implementation of such two prototypes is definitely non-trivial, which I really appreciate the authors' effort. While overall, the manuscript is in a reasonably good shape in terms of concept, writing, technical flow, and exposition, there are several comments/concerns that the authors should clarify or elaborate so as to lead to a more convincing scientific article.

1. To my best knowledge (with experimental observation and insights from several industrial players in the optics combiner field), waveguides, no matter volume hologram type or surface-relief grating type, would show noticeable impact on the polarization status of the input wavefront. This effect is often a bit complicated and hard to be disentangled since many parts/components could contribute to it. That said, the wavefront coming out from the waveguide and continuing to propagate to the wavefront cameras is no longer accurately sharing the same polarization as the reference beam. I believe adding a polarizer right after the waveguide out-coupler (in front of the cameras) would be an easy but crucial way to lead to a much higher SNR wavefront calibration. I am curious why the authors didn't do this. Even the authors claimed in the supplement that the polarization was ignored reasonably, it would be necessary to present a comparison with/without this extra polarizer.

2. Figure 2 seems very insightful. I would be curious why the authors would set up so many different Q and R kernels? Frankly, Q represents the modulation that happens on the in-coupler end? If that's the case, shouldn't it be the same for all channels which seems to be more physics-inspired and saving computing memory/resource? The same concern applies to the out-coupler end. Have the authors tried a different setting of the this multi-channel configuration, like 1-N-1 or 1-N-N? It would be really helpful to see a more solid justification on the multi-channel convolution model representation, since this is really the most interesting contribution.

3. The 3D results (Figure 6 and 7) unfortunately suffer from the poor quality, in terms of both resolution and defocus blur effect. Although I fully understand this could be very challenging, elaborated discussion could aid to the argument. Yet, as holographic displays are

essentially 3D-driven, more 3D results should be included in the paper. Now the entire manuscript with over 16 pages has only one set of results regarding 3D, or more rigorously speaking, 2.5D.

4. Following the above comment, considering the current image quality and the lack of natural 3D behavior, the title sounds a bit over-claimed and too general. First of all, "waveguide holography" isn't an entire novel concept. Many prior work, including several cited papers, for example [Yeom 2015; Jan 2018; Choi 2021], has already discussed the combination of waveguides and holographic display engines. In addition, the second part of the title "Towards True 3D Holographic Glasses" holds for almost every research paper that was recently published or is to be published in this field. As such, for a scientific article (not a PR story-telling article) on such a high-impact journal, I would assume a more specific, appropriate title should be made. In the meanwhile, I would suggest rewording the tone accordingly across the introduction and abstract to sound more humble and solid.

5. The authors claim to retain the resolution advantage of holographic displays. However, there is no results of standard resolution charts or patterns being presented. I would suggest adding the comparison with/without the proposed model/calibration using a resolution chart target. This would be insightful in convincingly showing not only resolution but contrast performance. Both of these would be highly significant for AR glasses.

6. Regarding the super-bright spot in the center of almost all results, is it really just the DC term issue of the SLM? Was Fresnel holography or Fourier holography being used? Or is there any practical trick to partially mitigate it in the generation of holograms? I believe this issue matters significantly for near-eye displays. Please clarify in the manuscript.

7. The termed "waveguide design" part seems a bit brute-forced. Basically, the authors still rely on existing grating equation and cherry-pick a thickness? Since the authors were fabricating their own customized waveguide anyways, it would be great to note the discussion of optimizing a waveguide in a more deterministic manner, including but not limited to thickness, refractive index, grating parameters, etc. Please clarify.

8. How well would the wavefront calibration and the multi-channel model be generalized to different types of diffractive waveguides?

9. Do the authors have an idea if it could actually be fine to remove the high-cost and maybe ambiguous wavefront camera settings here? Instead, how good the results would be if just using conventional intensity sensors to supervise the model learning?

10. It would be beneficial to have the actual photographs and layouts of the two prototypes in the suppl. at least, rather than just schematic diagrams.

11. I appreciate the detailed elaboration of future possibilities in Figure 10, although very interesting, this would not be justified as the core contribution of current manuscript. As such, maybe shorten these paragraphs a bit in the main text and yield the detailed elaboration to the supplement?

12. I am also curious how would recent emergence of AI/DNN fit into

this work, considering such an amazing amount of AI-driven holographic display solutions have been recently launched in academia?

===== Author's Response to Reviewer #1 (point-by-point) =====

1. To my best knowledge (with experimental observation and insights from several industrial players in the optics combiner field), waveguides, no matter volume hologram type or surface-relief grating type, would show noticeable impact on the polarization status of the input wavefront. This effect is often a bit complicated and hard to be disentangled since many parts/components could contribute to it. That said, the wavefront coming out from the waveguide and continuing to propagate to the wavefront cameras is no longer accurately sharing the same polarization as the reference beam. I believe adding a polarizer right after the waveguide out-coupler (in front of the cameras) would be an easy but crucial way to lead to a much higher SNR wavefront calibration. I am curious why the authors didn't do this. Even the authors claimed in the supplement that the polarization was ignored reasonably, it would be necessary to present a comparison with/without this extra polarizer.

We appreciate the reviewer's insightful comments regarding the polarization state change during waveguide propagation, and it is true that incorporating a polarizer at the out-coupler offers benefits in terms of calibration. In fact, we did place a polarizer at the out-coupler of the waveguide as the reviewer suggested, and we have mentioned this in Supplementary Material S1 as "*...Lastly, polarization of light is not considered since the SLM is active for only a single polarization, and we put a polarizer at the output side.*" We believe this also addresses the reviewer's concern about ignoring the orthogonal polarization.

However, we recognize that we could make this point clearer in the manuscript to prevent misunderstandings. We have revised the manuscript in the "Architecture of waveguide holography" section:

"The system consists of a collimated laser light source, a spatial light modulator (SLM), polarizers, and a pupil replicating waveguide with surface relief gratings."

"The system consists of a collimated laser light source, a spatial light modulator (SLM), a pupil replicating waveguide with surface relief gratings, and linear polarizers laminated on the SLM and out-coupler of the waveguide."

Additionally, in the Supplementary Material, we have revised as:

"...Lastly, polarization of light is not considered since SLM is active to only a single polarization and we put a linear polarizer at the out-coupler of the waveguide, matching the SLM polarization"

direction. Some light is lost at the out-coupler due to the polarization state changes inside the waveguide, however it improves the contrast of the hologram.”

2. Figure 2 seems very insightful. I would be curious why the authors would set up so many different Q and R kernels? Frankly, Q represents the modulation that happens on the in-coupler end? If that's the case, shouldn't it be the same for all channels which seems to be more physics-inspired and saving computing memory/resource? The same concern applies to the out-coupler end. Have the authors tried a different setting of the this multi-channel configuration, like 1-N-1 or 1-N-N? It would be really helpful to see a more solid justification on the multi-channel convolution model representation, since this is really the most interesting contribution.

We greatly appreciate the reviewer's high regard for our waveguide modeling and their perceptive comments on it. As the reviewer points out, this is one of the core contributions of our work, and we are happy to discuss further details.

The complex-valued aperture Q is applied to the input field generated by the spatial light modulator and interacts with the waveguide in-coupler. Q models the wavefront modulation applied to the first-order diffraction term at the in-coupler plane, for example, input-aperture boundaries, diffraction efficiency, aberration, and impurities or defects on the in-coupler. Importantly, Q also selects the propagation "path" inside the waveguide according to the position where each photon lands. For instance, assume two different points A and B at the input-coupler and trace the path of waveguide propagation from these points. If the waveguide is homogeneous, they should propagate along the same path, and the system can be modeled with a single convolution kernel as described in the **Modeling of waveguide holography** section. However, in practice, the system is not homogeneous, and the output wavefront from A would not be exactly the same as B, which means the convolution kernel that models the waveguide propagation would differ as well.

Therefore, Q provides the degree of freedom to select or generate "customized propagation paths" by linear combination of convolution kernels. In other words, the multi-channel kernels h serve as the basis of waveguide propagation, while Q functions as a complex number coefficient for the basis kernels. Each point at the input-coupler would have a different linear combination of basis kernels, allowing the model to obtain spatial invariance. Simultaneously, linear summation induces a smooth transition between adjacent positions to prevent the model from becoming overly sensitive (if points A and B are close enough, they likely have similar paths), while still maintaining differentiable properties. Note that the system is not exactly linear summation since R is applied last, which models the modulation at the out-coupler similarly to Q and adds further freedom for phase matching.

The reviewer gave perceptive comments suggesting to explore different model structures. In fact, our modeling is the result of various experimental configurations of model structure, and we are glad to provide the results as follows:

(#ofQchannels-#ofhchannels-#ofRchannels)
(1-1-1): This case is the single channel case.

(N-N-N): Our model, $N=9$ (PSNR = 27.22dB, cPSNR = 21.28dB)

(1-N-1): This is essentially same as (1-1-1) since the N kernels can be merged into a single kernel as a complex summation of them, using the linear property of convolution.

(1-N-N): In this case, all the kernels h share the same input complex aperture, therefore the model loses the capacity to handle the spatial-variant property which is described above. As a result, the fidelity of prediction drops significantly (PSNR = 23.39dB, cPSNR = 17.48dB).

(N-N-1): In this case, all the kernels h share the same output complex aperture R . We still see a slight drop of PSNR, however not as drastic as (1-N-N) case. This result aligns with our physical intuition of the modeling, because R does not contribute a lot on the non-LSI characteristics. R is expected to capture the wavefront modulation at the out-coupler side or outside of the waveguide (e.g. polarizer), which we expect to show more homogeneous response than inner waveguide interactions. (PSNR = 25.91dB, cPSNR = 19.97dB)

As we believe that this discussion raised by the reviewer could be interesting for the readers who try to replicate our work, we decided to add the results in the **Ablation Study** section in the Manuscript:

“We also tested the contribution of each multi-channel parameter Q , h , and R . When Q is set as a single channel (1-N-N), all the kernels h share the same input complex aperture, therefore the model loses the capacity to handle the spatially variant property which is described above. As a result, the fidelity of prediction drops significantly as shown in Fig.3c. Meanwhile, when R channel is set as a single channel (N-N-1), the PSNR drop is relatively slight. This result aligns with the physical intuition of the modeling as R is expected to capture the wavefront modulation at the out-coupler side or outside of the waveguide, which we expect to show more homogeneous response than inner waveguide interactions.”

We have updated the Figure 4 (previous version) and merged with Figure 3 (revised version) and updated the figure caption accordingly.

3. The 3D results (Figure 6 and 7) unfortunately suffer from the poor quality, in terms of both resolution and defocus blur effect. Although I fully understand this could be very challenging, elaborated discussion could aid to the argument. Yet, as holographic displays are essentially 3D-driven, more 3D results should be included in the paper. Now the entire manuscript with over 16 pages has only one set of results regarding 3D, or more rigorously speaking, 2.5D.

We appreciate the reviewer’s constructive feedback suggesting to provide more 3D results. We are aware that some recent research have demonstrated 3D holographic images with photorealistic blur using a several different methods (e.g. multi-layers, focal stack, hogel-based or lightfield-based optimization, etc.). Improving on CGH rendering for more natural blur, occlusion, and sometimes texture is an important research topic that we are interested in as well. However, we did not put too much effort for 3D hologram rendering in this work because our core contribution is not the algorithm to generate high quality CGH, and we believe the capability to display 3D image is now widely accepted for holographic display communities. Since our previous 3D result (fish and dandelion image) rendering method does not force the “shape of the blur” explicitly, defocus effect may not look natural as reviewer has pointed out. However, the in-focus effect and size of the blur are correctly reproduced, which we believe can be a good evidence to prove the capability as 3D displays.

Although, we appreciate the reviewer's feedback and have added new 3D results with more realistic blur. We used focal-stack target and used time-multiplexing CGH optimization method to create the CGH, without making change in our waveguide holography pipeline. We used a focal stack target of 12 planes with accurately rendered blur and occlusion. Although our waveguide is optimized/fabricated only for a single wavelength, we demonstrated the pseudo color images by merging RGB channel of the image in order to facilitate the intuitive visual perception of the 3D effect and image quality. The number of frames were selected based on recent studies on 3D CGH rendering. We believe the newly added results could improve the quality of our submission.

Revisions made:

Added focal stack result in Figure 5 (revised version) and revised the manuscript as follows:

In *Results*,

"Figure 5c shows the temporally-multiplexed 3D results captured with the wavefront camera. CGH is rendered using focal stack target (12 planes) with accurately rendered blur and occlusion. Since our waveguide is designed for a single wavelength, we showcase pseudo-color images by merging separately captured RGB channels images in order to facilitate the intuitive visual perception of the 3D effect and image quality."

As we have added more images, we have moved some of the previous results (Different exit-pupil results) images in the supplement and removed the mid-focus image of fish and dandelion result.

Also, we have updated Figure 2 (Illustration of waveguide holography model pipeline) by adding the CGH rendering pipeline to help understanding.

4. Following the above comment, considering the current image quality and the lack of natural 3D behavior, the title sounds a bit over-claimed and too general. First of all, "waveguide holography" isn't an entire novel concept. Many prior work, including several cited papers, for example [Yeom 2015; Jan 2018; Choi 2021], has already discussed the combination of waveguides and holographic display engines. In addition, the second part of the title "Towards True 3D Holographic Glasses" holds for almost every research paper that was recently published or is to be published in this field. As such, for a scientific article (not a PR story-telling article) on such a high-impact journal, I would assume a more specific, appropriate title should be made. In the meanwhile, I would suggest rewording the tone accordingly across the introduction and abstract to sound more humble and solid.

We concur that revising the title to be more specific could enhance our paper. However, we would like to re-emphasize the significant differences between our work and previous studies. Please refer to our response to Reviewer #1, especially category e):

Holographic displays with lightguide slab (non-pupil replicating waveguides). This category is the most relevant research that we could find, even though the titles and terminologies are

sometimes dissimilar. We have cited several papers [41-44], as these papers share a similar motivation to transmit holograms to the user's eye via waveguide. However, there are several important differences in the methods, as we have clearly stated in the manuscript. First and foremost, previous research uses non-pupil-replicating waveguides. In such cases, the architecture is similar to bird-bath type combiners with a diffractive optical element because there is only a single optical path and exit pupil that delivers the image to the user's eye. To ensure this condition, the overlap between wavefronts should be avoided in the waveguide propagation, which imposes some fundamental limitations on scalability: the eyebox position is fixed and small, and the field of view is also limited. Kress and Chatterjee also discuss this in their review article (Kress, Bernard C., and Ishan Chatterjee. "Waveguide combiners for mixed reality headsets: a nanophotonics design perspective." *Nanophotonics* 10, no. 1 (2021): 41-74) as:

"...The image over the input pupil can, however, be located in the near field when no pupil replication scheme is performed in the guide, ... (yielding a small FOV and small EB)"

Additionally, such designs require very thick slabs (e.g., 3 – 8 mm) compared to pupil-replicating waveguides (typically ~1 mm). We would like to emphasize again that thickness is one of the most important aspects characterizing the form factor of AR glasses. Often, these thick slabs are referred to as "lightguides" to distinguish them from pupil-replicating waveguides. The compact form factor is one of our main contributions, as well as the etendue/eyebox expansion capability of thin pupil-replicating waveguides.

We have cited 4 relevant studies and stated the differences in **Holographic displays using waveguide** subsection:

[Line 56] "However, there has been a fundamental limitation on scalability because the method is not intended to support pupil replication. The light guiding slab needs to be thick enough to avoid the replication, or the overlapped wavefront is scrambled, creating severe artifacts such as multiple ghost images and low contrast. Consequently, eyebox and field of view are limited to be small in such architectures."

[Line 61] "Our approach is fundamentally different from previous works in that pupil expansion is enabled to provide large eyebox with about a millimeter thick waveguide. The core idea of waveguide holography is to model the coherent light interaction inside the pupil replicating waveguide..."

Revisions made:

As suggested, we have revised our title in other to highlight our main contribution more specifically, and also clarify major differences from previous studies:

"Waveguide Holography: Towards True 3D Holographic Glasses"

"Waveguide Holography: Hologram Propagation via Pupil-Replicating Waveguides"

Previously, the focus was intended to emphasize the capability of displaying 3D imagery and the compact form factor that could be achieved. However, it did not clearly convey the core idea, and it could be confused with previous works based on non-pupil replicating waveguides. In the context of the revised title, "hologram propagation" refers to the process by which the reconstructed wavefront travels or propagates through the waveguide, and how this propagation can be modulated or controlled using pupil-replicating waveguides. We believe our new title is more humble and suitable for a high-impact scientific article, as the reviewer suggested.

Abstract has been revised to state the contributions more specifically as follows:

"We introduce a new near-eye display concept that incorporates a waveguide combiner, a spatial light modulator, and a laser light source. We demonstrate controlling the output wavefront of a waveguide by modeling the coherent light interactions inside the waveguide combiner and modulating the input wavefront using a spatial light modulator. The proposed method enables the display of full depth range 3D holographic images via a pupil-replicating waveguide with a large software-steerable eyebox. The method also offers additional advantages such as compact form factor realized with a lens-less holographic projector, and resolution enhancement capability by suppressing phase discontinuities from pupil replication. We build prototypes to verify the concept with experimental results and conclude the paper with summary and discussion. The combination of the two state-of-art display technologies; waveguide image combiner and holographic displays, unlocks new potentials towards true 3D holographic augmented reality glasses."

Please refer our response to Reviewer #1 for our revisions on Introductions.

5. The authors claim to retain the resolution advantage of holographic displays. However, there is no results of standard resolution charts or patterns being presented. I would suggest adding the comparison with/without the proposed model/calibration using a resolution chart target. This would be insightful in convincingly showing not only resolution but contrast performance. Both of these would be highly significant for AR glasses.

We agree on the reviewer's point that resolution and contrast are important factors in evaluating display performance, and we have attempted to provide as much proof as possible to demonstrate this. We present comparisons of image quality with and without our method in Fig. 5 (previous version) and the **Supplementary Material**, and we believe the (qualitative) difference is clearly demonstrated.

However, we realize that providing a quantitative metric would offer even more compelling evidence. A resolution chart is one widely used pattern for evaluating imaging systems, which shows the contrast response according to several spatial frequencies. However, it only demonstrates a few sampled frequencies with limited directions (only horizontal and vertical), and converting it to the contrast could be heuristic. Therefore, we believe that measuring the Modulation Transfer Function (MTF) is a more comprehensive evaluation method that provides contrast for all the frequency components in all the

directions. We could directly measure and retrieve the MTF using our complex wavefront capturing camera, which is a well-defined metrology method in phase-shifting holographic microscopy.

As a result, we have added the MTF plot from the wavefront measurement in **Fig. 6**. While the Amplitude Transfer Function (ATF) is sufficient for describing coherent illumination situations, we believe providing the MTF would also be useful for the following reasons: 1) MTF is more widely used for display quality evaluation than ATF, and 2) ATF does not hold for quasi-monochromatic light, such as in temporal multiplexing situations, which is the case with our newly added 3D result. The MTF plot clearly demonstrates the contrast enhancement when comparing without vs with our method, in terms of intensity and cut-off frequency. It also shows the asymmetric contrast response according to the frequency component. Furthermore, we have added an inset in **Fig. 4** to facilitate the evaluation of resolution. The figure caption and manuscript has been updated accordingly.

6. Regarding the super-bright spot in the center of almost all results, is it really just the DC term issue of the SLM? Was Fresnel holography or Fourier holography being used? Or is there any practical trick to partially mitigate it in the generation of holograms? I believe this issue matters significantly for near-eye displays. Please clarify in the manuscript.

We use all the range between Fresnel - Fourier hologram, as our reconstruction depth varies 0 to infinity, but most of the depth range is close to Fourier hologram. When the image depth is infinity (0 diopter), its hologram is in Fourier domain and the DC noise from the SLM (where fill-factor is not 100%) forms a bright spot. We can observe that the shape of the bright spot becomes larger and eventually spread to the entire FOV in 0 depth. In simulation, it is possible to reduce the DC noise by optimizing the input hologram assuming 10% of DC energy. However, in the experiment, it is difficult to remove the DC noise completely since all the energy is focused in a single point, especially with the infinity depth.

Here we note our perspective on this issue as follows:

First, DC term in Fourier domain is not originating from our specific architecture or method, but it has existed for all Fourier type holograms using LCoS SLMs. Typically, researchers eliminate the DC noise using a half band filter in the relay system, by sacrificing the half of the SLM bandwidth, which we could have used as well. However, we chose not to use it since we wanted to avoid the further loss of SLM bandwidth (note that we only use partial area of the SLM), and also to focus on demonstrating the compact form factor.

As we noted in the Discussion section (previous version), there are some potential solutions other than half band filtering. First, we can tilt the SLM with respect of the waveguide so that the FOV is shifted and the DC noise is not coupled to the waveguide. An angular stop (notch) filter can be used for complete block. In theory, there is a potential to use discarded half bandwidth by re-mapping the input-output FOV (For example, $[-\theta : \theta] \rightarrow [0 : \theta] + [-\theta : 0]$), using the volume gratings with high k -vector selectivity. Alternatively, the DC noise can be filtered using an aperture at the in-coupler of the waveguide by modifying the hologram projection module with adding the projection lens, so the state of Fourier-Fresnel hologram is inverted.

Based on above discussion, we have added more detailed discussion with new illustrations in the **Supplementary Material** and have revised as follows:

In **Discussion**

"...We discuss related details including further miniaturization strategies and potential solutions for the DC noise in the Supplementary Material."

In **S5.3 Additional Notes on the Architecture**

"Due to the limited fill-factor of SLM, a small portion of light is reflected without modulated and generates a DC noise. When a Fourier hologram is displayed, the DC noise forms a bright spot at the center field of view. Typically, a half band filter is placed in the Fourier domain to eliminate the noise. To retain a compact form factor, an angular stop filters can be used without 4-f relay optics as shown in Fig. S9b..."

...For example, a projection lens can be used similarly to a conventional image projection module as shown in Fig. S9c, so the FOV is limited by the focal length of the lens and the size of the SLM. In this case, the DC noise is spread in the entire FOV domain. However, such configuration could focus DC noise close to the eye-box domain, that will result non-uniform eye-box quality. In this case, an input-coupler aperture or additional spatial filter can be used to filter the high-order and DC noise as shown in Fig.S9c. Nevertheless, we choose to showcase the feasibility of the ultimate lens-free architecture, betting on the future breakthrough in the micro display technology."

Also, we have added **Fig. S8b** and **c**.

7. The termed "waveguide design" part seems a bit brute-forced. Basically, the authors still rely on existing grating equation and cherry-pick a thickness? Since the authors were fabricating their own customized waveguide anyways, it would be great to note the discussion of optimizing a waveguide in a more deterministic manner, including but not limited to thickness, refractive index, grating parameters, etc. Please clarify.

As reviewer's comment implies, there are plethora aspects to consider in the design process of the waveguide. For convenience, we can divide the design parameters into three important groups as follows: layout of the waveguide (shape, size, and position of each grating), substrate selection (refractive index and thickness), and grating parameters (grating pitch and structure of surface relief grating). Combining all these parameters, the waveguide design becomes a very high degree of freedom task with multi-dimensional design space and trade-off relations. Thus, we put a lot of effort to explore the large design space and to narrow down the parameters for optimized performance, form factor, and practical fabrication. Although, we realize that our concise description may look overly simplified, and the design choice may not seem clearly justified. Therefore, we have revised our paper to discuss the waveguide design more comprehensively. Before discussing the detailed changes made, we'd like to start by summarizing the design process of each aspect including layout, substrate selection, and grating parameters.

Summary of the design process:

For the layout, we adopted the design introduced by Tapani Levola, “*Diffraction optics for virtual reality displays*,” *Journal of the Society for Information Display* 14 (2006), which consists of in-coupler, exit-pupil expanding grating (EPE grating), and out-coupler. Since introduced, this design is now widely used in industry and academia. The out-coupler size is designed first to cover the target eyebox size of 16 x 14 mm, which is considered large enough for near-eye displays. Then the trapezoid shape of EPE coupler is designed to deliver the target field of view efficiently, based on the simulated ray trajectory. We set the input coupler size comparable to user’s pupil size, to match the information amount of the input and target output pupil. The distances between each grating are set as compactly as possible because of following reasons. First, less propagation prevents potential artifacts such as scattering and photon loss from defects and impurities in the waveguide substrate. And second, smaller layout size is advantageous for achieving compact glass form factor.

For the waveguide substrate, refractive index and thickness should be selected. Thickness is an important parameter that affects the total-internal-reflection distance and thus the number and density of pupil replication. Additionally, it directly affects the thickness of the device, which is crucial to ergonomics and aesthetic aspect. Generally, having a high refractive index is desired because it results the larger TIR bandwidth (the range of possible modes that can be transmitted via total internal reflection), as well as it reduces unwanted artifacts such as visible diffracted light from world side. However, the refractive index cannot be arbitrarily selected in practical because of the limited choice of the material and price. Also, the fabrication method imposes limitations as well (in our case, nano-imprinting). For such reasons, we chose to use a typical glass material ($n=1.5$) in our work.

For the gratings, pitch and grating structures such as aspect ratio and shape need to be designed. The grating pitch decides the pupil replication distance and thus affects density of pupil replication along with the thickness of the substrate. Once the in-coupler grating is set, the summation of grating vector is designed to be 0, as described in **Supplementary Material S1**. The grating structure affects the efficiency and selectivity of diffraction on the wavelength and incident angle of light. For finetuning the structure, rigorous coupled wave analysis (RCWA) or finite-difference time-domain method is used. As the general goal is similar to surface relief grating design process for waveguide displays that is already introduced in academia, we do not elaborate it in the manuscript.

In the previous version, we presented summarized description of the design process in **Design space and scalability analysis** subsection and **Method - Waveguide fabrication** subsection in the manuscript as follows.

In **Design space and scalability analysis** subsection:

[Line 215] “The design of our waveguide shares general goals of conventional pupil replicating waveguides, such as high image resolution and high light throughput efficiency, as well as the uniformity of out-coupled light intensity in both eyebox domain and field of view domain.”

And in **Method – Waveguide fabrication** subsection,

[Line 284] “The thickness of the substrate is selected based on the simulation results presented in Fig. 9 (b) to achieve a good performance and still retain the thin form factor. The center TIR angle is set as 50 degrees with the center wavelength of 532nm wavelength. The waveguide is designed to support 20 degrees of horizontal field of view and 16 X 12 mm of eyebox size. The waveguide samples are fabricated using nano-imprinting method which is suitable for mass

production. The specifications of surface relief gratings such as shape, slant angle, and aspect ratio are fine-tuned using rigorous simulation to achieve spatial and angular uniformity at the eyebox domain. In general, targeting higher uniformity reduces the grating efficiency and thus trades overall efficiency. We set the merit function to balance between uniformity and efficiency, to achieve over 5% of end-to-end throughput efficiency in average and maximize the uniformity.”

Especially, among the various parameters involved in the design space, we identified a meaningful relation between pupil replication distance and the fidelity of CGH rendering as presented in *Design space and scalability analysis* subsection as we stated in the previous manuscript:

[Line 215] “we focus on pupil replication distance d_{rep} which is a function of thickness t and the TIR angle of corresponding field of view component...”

Also, in the following paragraph, we provide the physical intuition of the relation as:

[Line 216] “If d_{rep} is set too large, some field of view component may not fill the eye pupil and cause vignetting artifact or partial loss of the image. ... If the wavefront is replicated and overlapped too densely, it reduces the degree of freedom to control the interference in a desired manner, which trades the image quality.”

Note that d_{rep} is a function of thickness, grating pitch, refractive index, and wavelength. However, we did not provide the analysis for all three parameters to avoid the section to be too lengthy, and also the multi-dimensional parameter sweep is computationally prohibited. Instead, we reduced the dimension of analysis by setting the thickness as a main variable because the refractive index cannot be freely selected as described above, and it is tied together with the grating pitch to determine the central TIR angle. The result is presented in **Fig. 9b**. The analysis suggests the PSNR of CGH rendering fidelity peaks near 1.X mm of waveguide thickness, with refractive and grating pitch we used. This intuition is used as a rule of thumb to design the waveguide substrate and grating pitch as we stated:

[Line 216] “The plot suggests that there is a range of sweet spot that achieves the best image quality, where d_{rep} is balanced between pupil density and the wavefront optimization freedom.”

However, the above analysis is not a sole standard as the thickness is also an important parameter for form factor, weight, and cost aspects as described above. For display application, PSNR over 40dB is already considered high enough, and it does not show a practical difference compared with 100dB. Therefore, we select thickness to be 1.15 mm, where the form factor is minimized with achieving good enough PSNR (60dB).

Revisions that we made:

Although the detailed design process of waveguide is out of our scope and could be too lengthy, we agree with reviewer’s opinion in that the paper would benefit from providing detailed guidelines with physical intuition. To address the reviewer’s concern, we have made following updates.

First, we provide more general descriptions on the waveguide design process. And as the section became too lengthy, we decided to move the analysis part to the **Supplementary Material**. We believe

this new arrangement makes sense as the waveguide design would not be the most interesting part for most of the readers, yet to provide useful information for those who are willing to replicate our work:

In *Design space and scalability analysis* of *Supplementary Material*,

“Among the various parameters involved in the design space, we focus on pupil replication distance d_{rep} which is a function of thickness t and the TIR angle θ_{TIR} of corresponding field of view component as:

$$d_{rep} = 2 t \tan \theta_{TIR}.$$

If d_{rep} is set too large, some field of view components may not fill the eye pupil and cause vignetting artifact or partial loss of the image. Otherwise, when the pupil replication distance is too dense, the intensity of guided light decays too fast because it requires too many TIRs for the light propagation over the entire eyebox. In the conventional waveguide displays, the resolution tends to be degraded with thinner waveguide as effective numerical aperture is reduced and more clipping happens. In waveguide holography, the numerical aperture is not necessarily degraded as demonstrated in the previous section. However, it imposes a different restriction in the wavefront optimization. If the wavefront is replicated and overlapped too densely, it reduces the degree of freedom to control the interference in a desired manner, which trades the image quality. We can define the replication density ρ and degree of freedom for each k -vector (or FOV) component σ as follows:

$$\rho = S_{in}/d_{rep}$$

$$\sigma = d_{rep}/S_{in}$$

where S_{in} is the smaller of the exit pupil size of the projector or entrance pupil size of the waveguide. The absolute value of each does not necessarily predict the performance when it is close to or larger than 1, but if the value become too small, it is expected to negatively impact the image quality. The simulation results presented in Fig. S5a show that there is a range of sweet spot that achieves the good image quality, where d_{rep} is balanced between pupil density and the wavefront optimization freedom...

...Fig. S5b shows the scalability in terms of field of view varying the pixel pitch of the SLM. As the supported field of view increases, the maximum PSNR tends to decrease slightly; however, a PSNR over 40 dB would not make a practical difference in perceived image quality. Figure S5c illustrates both artifacts related to the thickness in simulation, using the specifications of the benchtop prototype. The perceptual artifact in the field of view domain is visualized in Fig. S5d, varying the system FOV. When the waveguide thickness is too thin, the artifact tends to be dependent on the texture of displayed image since the cause is limited optimization freedom. While the thickness is too thick, the artifact tends to show arbitrary pattern independent to the contents because the vignetting from the pupil density becomes the main limiting factor.

The simulation results provide valuable insights for optimizing the design parameters of the system, as well as demonstrating the architecture's scalability. In Fig. S7a, we plot the design

space of the waveguide guide based on its thickness and center TIR angle. The center TIR angle is selected based on the refractive index of the substrate and the target field of view, which is set to 50 degrees in our case. Based on the simulation results, our design strategy is to simultaneously maximize the PSNR and minimize the thickness of the waveguide...”

Also, we revised **Fig. S5** and we have performed simulation presented **Fig. S7a**.

Second, we elaborate on the physical implication of tradeoff relation on pupil replication distance, with thorough analysis and numerical demonstration. In the initial submission, the explanation remained to be high level and underlying physics is not conveyed clearly. In the revised version, we explain the degree of freeform for wavefront optimization in the guided-wave optics regime:

In *Design space and scalability analysis* of *Supplementary Material*,

“Physically, the degree of freedom for wavefront optimization can be interpreted in the guided-wave optics regime when d_{rep} is small. As described in Eq. 2 in the manuscript, the theoretical mode spacing of the waveguide is determined as $\Delta\theta_{res} = \lambda / 2 t \tan\{\theta_T\}$. In the ideal case of boundary/loss-less waveguide with an infinite input beam size, only discrete angular modes can be transmitted through the waveguide. However, in practical pupil-replicating waveguides with leaky gratings and physical boundaries, the transfer function of the waveguide becomes quasi-continuous with varying amplitude. When this effect is combined with discrete angular mode spacing of input wavefront generated from pixelated SLM, it can be predicted that some angular components of the input field will be lost during the waveguide propagation, as shown in Fig. S6. As the thickness becomes thinner and d_{rep} decreases, this signal loss will negatively affect the optimization freedom for the output wavefront. It could also explain the ringing in Fig. S5a as a result of Moiré effect between the transfer function of the waveguide and the discrete input field from the SLM. However, the typical mode spacing of waveguide is dense enough for human visual acuity, and some loss would be tolerable without noticeable degradation.”

Also, we have performed simulation presented **Fig. S6** and added figure caption as:

“Illustration of the mode spacing characteristics of pupil-replicating waveguides. Amplitude of the transfer function is calculated as a ratio of input and output energy transmitted through the pupil-replicating waveguide corresponding to its angular component, simulated based on section S1. Note that the peak of simulated transfer function matches the theoretical mode spacing of ideal waveguide, while both are simulated/calculated separately. The product of discrete angular components of input field from the SLM and the transfer function predicts the signal loss that affects the degree of freedom of wavefront optimization.”

Lastly, we will share the **physical simulation code** that we used to generate the simulation results, so that the readers can test with various parameters discussed above and even more, to explore the large design space.

8. How well would the wavefront calibration and the multi-channel model be generalized to different types of diffractive waveguides?

As described in **Modeling of waveguide holography** Section, our model is based on the assumption that the waveguide propagation consists of three linear interactions: optical propagation in the waveguide substrate, total internal reflection at the substrate boundary, and the first order diffraction at the gratings. We also present the analytic derivation and reasoning for our modeling in **Supplemental Material S1**. Provided that the waveguide propagation is based on the same interactions, we believe our calibration method and the model can be generalized to it.

As described in **Supplementary Material Section S1**, there are different types of diffractive waveguides based on types of diffractive coupler. For example, there are SRG waveguide, volume Bragg grating (VBG) waveguide and polarization volume hologram (PVH) waveguide, or hybrid version of different types of couplers. In the system level, the main difference among these waveguides is the material or nano-structure of the gratings, which results to different polarization characteristics or bandwidth/selectivity characteristics of the diffraction. However, we expect such differences would not be drastic in the Fourier optics regime, where our model is built based on. In specific, the difference in the polarization characteristics can be neglected as we used polarizer at the out-coupler while the difference in the bandwidth. Also, the difference in selectivity characteristics of the diffraction could be still captured in multi-channel kernels h .

9. Do the authors have an idea if it could actually be fine to remove the high-cost and maybe ambiguous wavefront camera settings here? Instead, how good the results would be if just using conventional intensity sensors to supervise the model learning?

We really appreciate the reviewer's question as we believe this is a very perceptive question that could verify our contribution of adopting wavefront camera. As reviewer suggested, we have tested our model training using the intensity only data, and the result showed drastic difference of performance. In short, the training only worked with the wavefront data, while intensity only data failed to train the model (cPSNR = 13.09dB, which is far worse than our single channel model, although PSNR was relatively high 23.44dB).

The result proves our claim in "Model calibration using complex wavefront camera" Section (previous version):

[Line 133]"...Compared with generic free space propagation, the light propagation inside waveguide results complicated overlapping and coherent interference that scrambles the amplitude and phase. By only measuring the amplitude, it is difficult to infer the waveguide kernels and complex apertures in the model. With wavefront camera, the access to phase information successfully retrieves the coherent light interaction in the waveguide."

We also believe that our method can be generalized to calibrate other types of holographic display systems with higher fidelity and efficiency, especially when the system involves complicated light transportation.

Additionally, we want to re-emphasize that this is the first to show one-time calibration for large 3D eyebox. Current state-of-the-art intensity-based calibration has a critical issue when the system has a large eyebox (which would be the practical situation). Once the system is calibrated at a certain exit-pupil size and position using conventional intensity-based methods [Ref. 21, 22, 59], the model is not guaranteed to show similar quality of calibration at different positions or different sizes of pupil (note that our intensity-only training result showed high PSNR, but fails to retrieve exact complex field). It would be very challenging to repeat the calibration for all possible four dimensions (x,y,z, size). Meanwhile, our method can directly predict the complex wavefront at the pupil plane, and thus it is possible to freely select different size and 3D position of the pupil aperture for hologram rendering at different position.

Lastly, we would like to note that there are some low-cost and simple methods to achieve wavefront retrieval without using piezo actuator. For example, off-axis holography does not require phase shifting, therefore the cost could be similar to intensity-based calibration methods. However, we choose to use phase shifting holography since the off-axis holography should trade the bandwidth in frequency domain as the information is cropped near the carrier frequency, decided by the off-axis angle.

10. It would be beneficial to have the actual photographs and layouts of the two prototypes in the suppl. at least, rather than just schematic diagrams.

We have included the photograph of benchtop prototype in the **Supplementary Material Fig. S8** as the reviewer suggested. We have already provided the photograph of our compact prototype in **Fig. 1** (previous version). We have added one more photograph of the compact prototype in **Supplemental Material Fig. S4**.

11. I appreciate the detailed elaboration of future possibilities in Figure 10, although very interesting, this would not be justified as the core contribution of current manuscript. As such, maybe shorten these paragraphs a bit in the main text and yield the detailed elaboration to the supplement?

We agree with the reviewer's suggestion. We have moved the section to **Supplementary Material** and revised the manuscript.

“Although, we note that there are alternative variations of hologram projection module to increase FOV immediately. For example, a projection lens can be used similarly to a conventional image projection module as shown in Fig.S9c, so the FOV is limited by the focal length of the lens and the size of the SLM. In this case, the DC noise is spread in the entire FOV domain. However,

such configuration could focus DC noise close to the eye-box domain, that will result non-uniform eye-box quality. In this case, an input-coupler aperture or additional spatial filter can be used to filter the high-order and DC noise as shown in Fig. S9c. Nevertheless, we choose to showcase the feasibility of the ultimate lens-free architecture, betting on the future breakthrough in the micro display technology.

For further miniaturization, the illumination module can be modified. One potential is to directly illuminate the SLM through the waveguide. However, if the input beam is illuminated through the SRG waveguide, the reverse path at the in-coupler will generate unwanted diffraction term that is mixed with the signal as shown in the left of Fig. S9d, indicated as -R1. As a solution, polarization volume grating in-couplers can be used to eliminate unwanted diffraction orders due to as shown in the right of Fig. S9d. Also, as shown in Fig. S9e, collimated illumination can be generated in a more compact manner using a single mode waveguide or a beam expanding wedge prism”

12. I am also curious how would recent emergence of AI/DNN fit into this work, considering such an amazing amount of AI-driven holographic display solutions have been recently launched in academia?

As the reviewer points out, some researchers such as Peng et al. [22], Choi et al [21, 59], and Chakravarthula et al. (Learned hardware-in-the-loop phase retrieval for holographic near-eye displays. ACM Trans. Graph. 39, 6, Article 186, 2020) have adopted AI/DNN for hologram rendering to boost the image quality. One typical approach is to insert UNETs in the forward model of wave propagation to increase the model capacitance and achieve better calibration.

Meanwhile, Shi et al. [20] demonstrated the CNN only forward wave propagation model without using UNET. Essentially, one of their core ideas that made it possible is to limit the receptive field of the successive convolution layers. They re-configure the geometry of display setup so that each propagation can be modeled by minimum kernel size (smaller point spread function). Their approach works well with limited depth range as the CNN layers efficiently model the wave propagation and enable fast framerate. However, we believe such method is not suitable for waveguide system (note that their benchtop’s optical path consists only with generic free-space propagations). In waveguides, the pupil replication process inherently enlarges the receptive field size to the output coupler size. Therefore, the receptive field size becomes much larger if we apply the same method. In a quick comparison, they were able to model the propagation using 30 layers of CNN layers with (3x3) kernel size, resulting 60x60 pixels of receptive field. In our case, our receptive field size always becomes the total output coupler size which is 4000 x 3000 pixels (or 16 mm x 12 mm) due to the pupil replication, resulting the area of receptive field more than 3 order of magnitude larger.

Also, in a sense, our method is already built based on the essence of the machine learning framework. Our model is differentiable, and we use backpropagation to train the model. Our codebase is written in PyTorch, which is widely used in various deep learning and AI applications. One notable difference is that we try to use physically inspired model parameters and intentionally avoid using non-linear layers such as ReLU activation, as we believe the real-world system primarily consists of linear operations, as

stated in the manuscript. Therefore, our model is both mathematically simple and physically interpretable compared to models that heavily rely on DNNs. This intuition can be very useful for system parameter design, as we have demonstrated in the **Supplementary Material**. Furthermore, our model offers unique advantages such as access to analytic gradients, as demonstrated in **Supplementary Material Section S2**.

Often, differentiable models provide a large freedom in terms of cost function engineering and sometimes it allows huge merits (e.g., to reduce the speckle or induce natural blur). In this context, we believe our method has an important novelty since we are first to use complex wavefront to evaluate the cost function, which can expand the freedom of cost function engineering to the complex number domain.

In fact, during the early stage of our research, we have tested to use UNET for waveguide modeling (with 2 input/output channels: real/imaginary part of complex wavefront) and compared the result with our waveguide model. And as a result, it simply did not work. The PSNR/cPSNR did not improve more than $\sim 22\text{dB}/\sim 15\text{dB}$, which was much worse than our single channel model, despite the fact that the model size was bigger. We believe our comment on the receptive field above could explain the result. There might be room for improvement such as finetuning the hyper parameters or adopting better models, however, we decided not to discuss this in the manuscript because of following reasons. First, this is not our main research direction that we pursue. When using UNET, physical layers in the forward model are intertwined with UNET and sometimes loses the physical meaning unless forced somehow, which deters our purpose of physically inspired modeling and compromises the fair comparison. And secondly, it would take lots of effort to thoroughly study all the possible combinations of model structures utilizing DNNs, which is also out of our main scope. Although, if the reviewer suggest it's necessary, we are willing to provide the above result in the bar plot and add some comments on it.

Reviewer #1 (Remarks to the Author):

I have re-reviewed this manuscript.

My original significant comment was concerned with the novelty of the paper. It was always clear that the technical detail was novel but I was concerned with the fact that the authors were, overall, claiming a new concept. They have considerably clarified this and changed the text accordingly. The nomenclature in the literature on this field is quite complex, but I believe they have now clarified this. Thank you.

So I am happy now to recommend publication.

Reviewer #2 (Remarks to the Author):

I highly appreciate that the authors have made quite an amount of revision/edit effort in this round, including conducting additional experiments, so that the readability and technical soundness have been largely improved, at least to my knowledge. However, after reading the updated version, in tandem with the response letter, there are still a few comments/concerns that, if resolved, would benefit general readers significantly.

1. As mentioned in the earlier review comment, I sort of believe demonstrated 3D results indeed matter for holographic displays, especially when involving a waveguide (which is mostly designed for 2D). In fact, modelling the propagation of a 3D complex field inside the waveguide is the key/challenging part of this work. Thus, I am still a bit surprised that the updated 3D results (Fig. 5) would still look quite poor even for single color. If I understand it correctly, these 3D results are obtained with focal stack supervision and time-multiplexing? Then why they still don't look closer to the state-of-the-art? Maybe it would be the visualization issue, but why such small FoV and such bad color fidelity? As NC aims for educating general readers in the field, seeking/demonstrating a reasonably good image quality would really be valuable.

2. Although the updated title reads much more appropriate than the original fancy high-level one, I would still highly suggest avoiding using "Waveguide Holography" this term too much. Technically, the second half of the updated title justifies itself already in terms of representing core technical contributions, so something like "Modelling Hologram Propagation via Pupil-Replicating Waveguides" would be quite attractive. In addition, as authors mentioned, there were already many prior papers using this term. Although there would always be differences, it doesn't bring pros to further confuse general readers just because of the desire of leveraging this fancy term for "PR" purposes. For such a high-impact journal, being humble doesn't hurt.

3. The authors claimed in the response letter that it's because the full depth range (Fourier + Fresnel) was used that the DC term (bright spot) would be inevitable and highly noticeable. I am a bit confused considering that many papers in the field in fact could use Fresnel holograms + lens (eyepiece), which is already there, to map the (virtual) depth range (on the SLM end) to a (real) full range (near to far) as well. That said, the DC term should be spreading out across the space or suppressed a bit by optimization. Then why not implementing this configuration or at least having a comparison test with state-of-the-art in the suppl.? Was it also due to the fact that Fourier holograms would be easier to optimize with a better visual quality result? The current version is not highly convincing. I think this part can benefit from an elaboration.

4. Wording can still use a polishment. For instance, the first sentence of abstract is still over-claimed, "We introduce a new near-eye display concept that incorporates a waveguide combiner, a spatial light modulator, and a laser light source..." Technically, there is not much new here in physically putting these three parts together (prior work has intensively investigated this display combination). The key of this work is, again, as authors mentioned in the letter, the new modelling of the hologram propagation along the waveguide. Another minor comment, the authors seem to use in too many places "... we are the first...". I would personally avoid addressing this too much in such high-impact article. As such, lowering the tone of statements across the full manuscript

would help.

=====

Author's Response of Paper Titled:

*Waveguide Holography: Modeling Hologram Propagation
via Pupil-Replicating Waveguides*

=====

Submission ID: NCOMMS-22-37509

Previous title: "Waveguide Holography: Hologram Propagation via Pupil-Replicating Waveguides"

Original title: "Waveguide Holography: Towards True 3D Holographic Glasses"

Original submission date: September 30th, 2022

1st Review received on: February 9th, 2023

2nd Review received on: May 22nd, 2023

Second revision submitted on: July 6th, 2023

REVIEWER COMMENTS

Reviewer #1 (Remarks to the Author):

I have re-reviewed this manuscript. My original significant comment was concerned with the novelty of the paper. It was always clear that the technical detail was novel but I was concerned with the the fact that the authors were, overall, claiming a new concept. They have considerable clarified this and changed the text accordingly. The nomenclature in the literature on this field is quite complex, but I believe they have now clarified this. Thank you.

So I am happy now to recommend publication.

===== Author's Response to Reviewer #1 =====

We appreciate the insightful second review and the reviewer's endorsement for publication. The reviewer's previous feedback significantly aided in improving the clarity and in detailing the novelty of our manuscript. We are pleased that our revisions met reviewer's expectations.

REVIEWER COMMENTS

Reviewer #2 (Remarks to the Author):

I highly appreciate that the authors have made quite an amount of revision/edit effort in this round, including conducting additional experiments, so that the readability and technical soundness have been largely improved, at least to my knowledge. However, after reading the updated version, in tandem with the response letter, there are still a few comments/concerns that, if resolved, would benefit general readers significantly.

1. As mentioned in the earlier review comment, I sort of believe demonstrated 3D results indeed matter for holographic displays, especially when involving a waveguide (which is mostly designed for 2D). In fact, modelling the propagation of a 3D complex field inside the waveguide is the key/challenging part of this work. Thus, I am still a bit surprised that the updated 3D results (Fig. 5) would still look quite poor even for single color. If I understand it correctly, these 3D results are obtained with focal stack supervision and time-multiplexing? Then why they still don't look closer to the state-of-the-art? Maybe it would be the visualization issue, but why such small FoV and such bad color fidelity? As NC aims for educating general readers in the field, seeking/demonstrating a reasonably good image quality would really be valuable.

2. Although the updated title reads much more appropriate than the original fancy high-level one, I would still highly suggest avoiding using "Waveguide Holography" this term too much. Technically, the second half of the updated title justifies itself already in terms of representing core technical contributions, so something like "Modelling Hologram Propagation via Pupil-Replicating Waveguides" would be quite attractive. In addition, as authors mentioned, there were already many prior papers using this term. Although there would always be differences, it doesn't bring pros to further confuse general readers just because of the desire of leveraging this fancy term for "PR" purposes. For such a high-impact journal, being humble doesn't hurt.

3. The authors claimed in the response letter that it's because the full depth range (Fourier + Fresnel) was used that the DC term (bright spot) would be inevitable and highly noticeable. I am a bit confused considering that many papers in the field in fact could use Fresnel holograms + lens (eyepiece), which is already there, to map the (virtual) depth range (on the SLM end) to a (real) full range (near to far) as well. That said, the DC term should be spreading out across the space or suppressed a bit by optimization. Then why not implementing this configuration or at least having a comparison test with state-of-the-art in the suppl.? Was it also due to the fact that Fourier holograms would be easier to optimize with a better visual quality result? The current version is not highly convincing. I think

this part can benefit from an elaboration.

4. Wording can still use a polishment. For instance, the first sentence of abstract is still over-claimed, "We introduce a new near-eye display concept that incorporates a waveguide combiner, a spatial light modulator, and a laser light source..." Technically, there is not much new here in physically putting these three parts together (prior work has intensively investigated this display combination). The key of this work is, again, as authors mentioned in the letter, the new modelling of the hologram propagation along the waveguide. Another minor comment, the authors seem to use in too many places "... we are the first...". I would personally avoid addressing this too much in such high-impact article. As such, lowering the tone of statements across the full manuscript would help.

===== Author's Response to Reviewer #2 (point-by-point) =====

1. As mentioned in the earlier review comment, I sort of believe demonstrated 3D results indeed matter for holographic displays, especially when involving a waveguide (which is mostly designed for 2D). In fact, modelling the propagation of a 3D complex field inside the waveguide is the key/challenging part of this work. Thus, I am still a bit surprised that the updated 3D results (Fig. 5) would still look quite poor even for single color. If I understand it correctly, these 3D results are obtained with focal stack supervision and time-multiplexing? Then why they still don't look closer to the state-of-the-art? Maybe it would be the visualization issue, but why such small FoV and such bad color fidelity? As NC aims for educating general readers in the field, seeking/demonstrating a reasonably good image quality would really be valuable.

First, we would like to thank the reviewer again for the suggestion in the initial review to add more 3D results. We believe our newly added results could enrich our contents and provide more intuitive visualizations to readers. In this review, the reviewer suggests that the quality of our 3D results may not be as good as other state-of-the-art results. Although, we think it is "apples to oranges" comparison, based on the following reasons:

- First of all, our experiment uses a pupil replicating waveguide image combiner. Most "state-of-the-art" results do not have restrictions in selecting the system parameters, and thus usually choose the simplest possible optical path (mostly, generic free-space propagation). In our system, the optical wavefront interacts with numerous optical surfaces, including TIR reflections and diffraction with three gratings, which generates tens of copies with aberrations and stray lights. Both calibration challenge of such a complex system and signal loss due to noise (e.g. stray lights, high order diffractions) should be taken into consideration when comparing the image quality. To the best of our knowledge, there is no fair comparison point in the previous

publications (even in Refs [47]-[50], they have much simpler optical path with non-pupil replicating waveguides, yet not as good as our results).

- Our model calibrates a much larger 3D eyebox area (up to 16 x 12 mm), with a single calibration process. Most of the state-of-the-art results that utilize DNN aided calibration show good calibration result with a pin-point eyebox, however, are not guaranteed to work in a different position or with different pupil size/shape with the same calibration result.
- The field of view is decided by the SLM pixel pitch as we use lens-less projection. Please refer our point-by-point response #3 for more details.
- Although we acknowledge that the color fidelity is not perfect, the source image is also not very vibrant color. We provide the target image in Fig.2 for the comparison.

However, we are not claiming that the improvement of the image quality is impossible. We accept fair criticism on our image qualities compared with other results, and we are very interested in exploring the methods to close the gap. Although, in this work, we are at the very early stage of exploring a relatively challenging optical system, and we believe the current results demonstrate reasonably good image quality for the proof of the concept. We would like to leave further improvement on the image quality as a natural next step for the future works.

2. Although the updated title reads much more appropriate than the original fancy high-level one, I would still highly suggest avoiding using "Waveguide Holography" this term too much. Technically, the second half of the updated title justifies itself already in terms of representing core technical contributions, so something like "Modelling Hologram Propagation via Pupil-Replicating Waveguides" would be quite attractive. In addition, as authors mentioned, there were already many prior papers using this term. Although there would always be differences, it doesn't bring pros to further confuse general readers just because of the desire of leveraging this fancy term for "PR" purposes. For such a high-impact journal, being humble doesn't hurt.

We are glad that the reviewer agrees our new title is much more appropriate than the original version. We are willing to seriously take further suggestions into account and reflect them. In this revision, the authors went through extensive discussion regarding the additional title change suggested by the reviewer. And we decided to not use the term too much as Reviewer suggested, but we wish to ask the reviewer's opinion again on keeping the "waveguide holography" only in the title, for the following reasons:

In fact, there are **only 2 academic papers** (with the same first author) that used this term according to our literature search, and the main contribution of these papers is the fabrication of the optical elements that use waveguide as an illumination source. As we mentioned in the previous response, such concept has been more frequently referred to as "waveguide hologram" in other (previous) papers. By definition, "hologram" refers to the interference pattern itself or fabricated optical elements.

Meanwhile, we believe “waveguide holography” could include a broader methodology that is described in our paper, such as modeling the hologram propagation, calibration method, hologram rendering pipeline, and system implementation.

Additionally, we believe the reviewer may be aware of some existing trends in academic community of this field, to title the paper in the format: “[XYZ] holography,” which we believe is not only beneficial for PR purpose, but is actually helpful to efficiently convey the core idea and facilitate a better understanding by readers. We believe our keyword that best summarizes the contributions is “waveguide.” Also, as we have already published our paper on Arxiv in September 2022, we are concerned that a drastic change in the title would confuse our fellow researchers who already recognize our paper by its initial title.

Therefore, we have updated our title again:

“Waveguide holography: Hologram Propagation via Pupil-Replicating Waveguides”

“Waveguide holography: Modeling Hologram Propagation via Pupil-Replicating Waveguides”

We think the last part now can precisely describe the core contribution, as well as differentiate our work from previous works to prevent confusion among general readers.

Also, we tried to refrain from using the term “waveguide holography” in the manuscript when it’s not absolutely necessary (**14 times  3 times**). Instead, we used more descriptive phrases to avoid potential confusion. We only used the term to refer to the title or proposed methods as a whole.

3. The authors claimed in the response letter that it's because the full depth range (Fourier + Fresnel) was used that the DC term (bright spot) would be inevitable and highly noticeable. I am a bit confused considering that many papers in the field in fact could use Fresnel holograms + lens (eyepiece), which is already there, to map the (virtual) depth range (on the SLM end) to a (real) full range (near to far) as well. That said, the DC term should be spreading out across the space or suppressed a bit by optimization. Then why not implementing this configuration or at least having a comparison test with state-of-the-art in the suppl.? Was it also due to the fact that Fourier holograms would be easier to optimize with a better visual quality result? The current version is not highly convincing. I think this part can benefit from an elaboration.

As the reviewer pointed out, use of a projection lens can remap the depth and the domain of the hologram (Fourier Vs. Fresnel) can be inverted. Using an eye-piece lens with an SLM positioned at the focal plane, depth range of the displayed image [0 diopter – infinite diopter] maps the hologram depth (or focus distance of kernel) into [0 mm – infinite distance]. This is just an inversion of domains; thus, the hologram should still cover Fourier + Fresnel domain to cover the literal “full depth range.” However, if

we limit the depth range to a practical focusable range (e.g. 0 diopter – 5 diopter), the hologram can mostly stay in Fresnel domain. Although, the reason we did not use the projection lens was a design choice considering the form-factor. For clarity, we itemize our response into several points and discuss each item.

Our design choice of lens-less projection module

As described in the **Architecture** section, this is our design choice to showcase the ultimate eye-wear-display form factor that can be achieved by the holographic display, by eliminating the projection lens and the optical propagation distance:

“Compared with the conventional waveguide display, the major difference is that the image projection module is replaced with the hologram projection module. The SLM is placed without any projection lens, eliminating the need of physical propagation distance, as well as achieving a light weight design.”

While some research papers on holographic displays focus on achieving high image qualities, sometimes physical constraints are not prioritized and design parameters (e.g. propagation distance, focal length of lens, etc.) are set to maximize the image quality. We focused on demonstrating miniaturization strategies that can greatly reduce the projector volume for the compact glasses type displays.

Other than the form factor, we note that each method has its own artifact in terms of DC noise. Our design choice results in the DC noise in the field of view domain. Due to the duality of Fourier-Fresnel domain, use of a projection lens does not eliminate the DC noise but sends them closer to the eye-box domain. Depending on the configuration, (e.g. placing the projection lens at the in-coupler plane and the focal length being similar to the propagation distance to the eyebox), the array of bright spot can be formed near the eye-box domain, which may increase the sensitivity to the pupil position error and degrade the eye-box uniformity especially when eyebox is large (similar effect is reported in here). In both cases, physical filtering can be an effective solution and we discuss for each case in the Supplementary:

“Due to the limited fill-factor of SLM, a small portion of light is reflected without modulated and generates a DC noise. When a Fourier hologram is displayed, the DC noise forms a bright spot at the center field of view. Typically, a half band filter is placed in the Fourier domain to eliminate the noise. To retain a compact form factor, an angular stop filters can be used without 4-f relay optics as shown in Fig. S9b.”

“...For example, a projection lens can be used similarly to a conventional image projection module as shown in Fig. S9c so the FOV is limited by the focal length of the lens and the size of the SLM. In this case, the DC noise is spread in the entire FOV domain. However, such configuration could focus DC noise close to the eye-box domain, that will result non-uniform eye-box quality. In this case, an input-coupler aperture or additional spatial filter can be used to filter the high-order and DC noise as shown in Fig. S9c. Nevertheless, we choose to showcase the feasibility of the ultimate lens-free architecture, betting on the future breakthrough in the micro display technology.”

Our emphasis on “full depth range”

We realize that our previous response may have led to some confusion. We apologize for any lack of clarity and would like to provide further explanation. Our mention of DC noise was not intended to suggest it as an inherent trade-off of achieving the full depth range. Instead, as we described above, it is related to our design choice of lens-less projection for a compact form factor.

We assume our emphasis on “full depth range” gave such an impression, but this is not specific to our design nor our novel contribution as it is not new in holographic displays. However, we presented this as an additional win that we can achieve using our complex wavefront calibration method that works well regardless of depth range (resulting in vastly different kernel sizes without a projection lens) with one-time calibration using complex wavefront.

Although, we realized that our previous emphasis on this feature may confuse the readers that we claim demonstration of a full depth range itself as a new contribution. To avoid such confusion, we decided to moderate our emphasis on this aspect as follows:

[Abstract]

“...The proposed method enables the display of full depth range 3D holographic images via a pupil-replicating waveguide with a large software-steerable eyebox.”

“...The proposed method enables the display of true 3D holographic images via a pupil-replicating waveguide, providing a large software-steerable eyebox.”

[Introduction]

“In the Results section, we experimentally verify the full depth range as well as étendue expansion, which enables a large 3D eyebox with a full native field of view of the spatial light modulator with software-steered eyebox.”

“In the Results section, we experimentally verify to display 3D images with the étendue expansion, which enables a large software-steered 3D eyebox.”

Fourier hologram Vs. Fresnel hologram?

This would be an interesting topic to compare the upper bound of image quality of Fourier Vs. Fresnel holograms. Unfortunately, we could not find any strong evidence in previous literature to compare the two. In theory, we believe there should be no major difference of upper bound of image quality of Fourier vs. Fresnel holograms assuming ideal SLM and light source. However, in our literature search, Fresnel type holograms are more frequently reported to demonstrate high image quality than Fourier type holograms. We assume this is at least partially related to the fringing-field effect of SLM. Fourier type hologram has more high frequency components by nature, which would be more vulnerable to such crosstalk. We carefully assume this is one of the reasons why some recent papers with high image quality tend to use Fresnel hologram, without explicit restrictions in choice of the configuration.

The reviewer has suggested a comparison between two configurations could be beneficial. From a modeling standpoint, we could simply modify the physical propagation module to incorporate the projection lens, without altering the waveguide model or calibration algorithm. However, executing such

a comparison experimentally isn't straightforward as it requires a major change in our experimental system, while the benefit would be somewhat out of the scope of our primary contributions. We have clarified our choice of implementation within the paper's logical structure, and thus, we don't see a comparison between alternative designs as essential. Nonetheless, we do find this an engaging topic and have endeavored to offer a high-level discussion on the design aspect in the supplementary materials, as indicated above. For now, we plan to leave a more thorough comparison for future work.

4. Wording can still use a polishment. For instance, the first sentence of abstract is still over-claimed, "We introduce a new near-eye display concept that incorporates a waveguide combiner, a spatial light modulator, and a laser light source..." Technically, there is not much new here in physically putting these three parts together (prior work has intensively investigated this display combination). The key of this work is, again, as authors mentioned in the letter, the new modelling of the hologram propagation along the waveguide. Another minor comment, the authors seem to use in too many places "... we are the first...". I would personally avoid addressing this too much in such high-impact article. As such, lowering the tone of statements across the full manuscript would help.

We appreciate the reviewer's constructive comments on writing. We did trim the wording in our first revision as there were some redundancies in claiming contributions. However, we do realize there is still more room to be polished. In this revision, we tried to refrain from over-emphasizing the contributions and lower the tone of statements, while delivering the scientific findings more clearly. We have also deleted some redundant sentences. We believe the current manuscript is now more appropriate for high-impact journal. Followings are the changes made:

"We introduce a new near-eye display concept that incorporates a pupil-replicating waveguide combiner, a spatial light modulator, and a laser light source."

"We introduce a holographic near-eye display concept that incorporates a pupil-replicating waveguide combiner, a spatial light modulator, and a laser light source."

"The combination of the two state-of-art display technologies; waveguide image combiner and holographic displays, unlocks new potentials towards true 3D holographic augmented reality glasses."

"The combination of the two state-of-art display technologies; waveguide image combiner and holographic displays, extends the potentials towards true 3D holographic augmented reality glasses."

"In this work, we propose a novel display concept that combines the advantages of both pupil replicating waveguides and holographic displays, enabling the path towards true 3D holographic AR glasses."

"In this work, we propose a display architecture that combines the advantages of both pupil replicating waveguides and holographic displays, enabling the path towards true 3D holographic AR glasses."

"We demonstrate that our method has the potential to overcome such resolution limitations."

"We demonstrate that such resolution limitations can be overcome by adoption of holographic displays, fully utilizing the coherent nature of light."

"The results suggest that our method opens up new possibilities for fully utilizing the coherent nature of light in waveguide displays."

[deleted]

"The core idea of is to model the coherent light interaction inside pupil-replicating waveguides. We introduce a novel multi-channel kernel modeling of waveguide wave propagation."

"The core idea of is to model the coherent light interaction inside pupil-replicating waveguides with a propagation with multi-channel kernels."

"The results verify that the focus spread artifacts are solved our model successfully solves focus spread artifacts and reconstructs the holograms via the waveguide even at the finite depth."

"The results verify that the focus spread artifacts are solved and holograms are reconstructed at desired depths via the waveguide."

"We demonstrate that our method has the potential to overcome such resolution limitations."

"We demonstrate that such resolution limitations can be overcome by adoption of holographic displays."

"Our method has successfully demonstrated reconstruction of 3D holographic images via a pupil-replicating waveguide with a large eyebox for the first time, unlocking new potentials for true 3D holographic augmented reality glasses."

[deleted]

Reviewer #2 (Remarks to the Author):

I have read through the very detailed response letter and briefly re-checked the revised manuscript. I would like to give authors credits for conducting such an amount of clarification effort. Those edits and clarifications have finally resolved most of my concerns.

I would also like to truly thank the authors for eventually accepting the comment and taking efforts in lowering the tone of contribution statements, although I do understand this might be a bit off from the "PR" perspective, especially witnessing the recent explosion of fancy papers in this research field. I really appreciate that.

While I would still suggest remove the fancy waveguide holography term in the title, this is not only due to the preference of a more humble tone as a scientific paper over a fancy tech report, but also because the waveguide could be a bit confusing since in optics there would be multiple types of waveguides, whereas this work only investigates one type of them (pupil-replicating). Nevertheless, I also hear and respect what the authors said in the letter. As such, I would leave it to the Nature's professional editor to make the call in terms of the title.

A humble yet sound tone for scientific papers is essential, and researchers/scientists should defend for it. In all, I would be fine with recommending the acceptance of this manuscript given the final version indeed incorporates all promised changes/edits.

Manuscript: NCOMMS-22-37509B

Dear Reviewer,

We are glad that our revision clarified most of the reviewer's concerns.

In the final review, the reviewer expressed concern:

"...While I would still suggest remove the fancy waveguide holography term in the title, this is not only due to the preference of a more humble tone as a scientific paper over a fancy tech report, but the waveguide could be a bit confusing since in optics there would be multiple types of waveguides, whereas this work only investigates one type of them (pupil-replicating)"

With respect, we humbly disagree with the reviewer's comment as described in our previous response letter, as we believe our revised title clearly and precisely describes the contributions without being overly broad or exaggerating. At its core, our research integrates waveguide displays with holographic displays, encompassing both algorithms, system architecture and experiment. Therefore, our title best summarizes the main idea, rather than being a fancy term just for PR purpose. To reviewer's concern on confusion, while there are multiple types/applications of waveguides, the fundamental principle remains consistent - a structure that guides light via total internal reflection. Especially, in the context of near-eye displays, this type of waveguide is the most relevant state-of-the-art technology, not merely "one type of them". Note that waveguide displays have already been productized in the market and the terminology is widely accepted in the industry and academia. Therefore, we believe there's little room for confusion.

Nevertheless, we want to respect the reviewer and editor's constructive suggestions. If this rebuttal does not address the concern, we are open to further discussion. As the reviewer suggested, we are communicating with the editor per final revision of our title, and we will do our best to follow editor's guidance and obey Nature publication's guidelines.

Finally, we sincerely appreciate the reviewer's time and enormous effort dedicated to our work. We believe reviewer's constructive comments helped tremendously to improve the quality of our manuscript.

Best regards,

Authors